# On Path Length, Beam Divergence, and Retroreflector Array Size in Open-Path FTIR Spectroscopy

Cameron E. N. Power[1,2], Aldona Wiacek[1,2]

[1] Department of Astronomy and Physics, Saint Mary's University, Halifax, B3H 3C3, Canada
[2] Department of Environmental Science, Saint Mary's University, Halifax, B3H 3C3, Canada

*Correspondence to*: Cameron Power (cameron.power@smu.ca)

**Abstract.** Open-Path Fourier Transform InfraRed (OP-FTIR) spectroscopy is an established technique used to measure boundary layer trace gas concentrations, consisting (in this work) of a spectrometer with an active mid-IR source coupled to a single transmitting and receiving telescope, and a cube-corner retroreflector array separated from the spectrometer and telescope by an atmospheric path. The detection limit at first increases with increasing optical path length (OPL) in the
atmosphere, which controls target gas spectral absorption depth; however, open-path beam divergence can lead to overfilling of the distant retroreflector array for separations greater than ~150 m (OPL ~300 m, details depend on specifics of spectrometer and telescope optics, plus array size), resulting in decreased returning radiation at the detector. In this case, the absorption signature of the target gas increases, but the signal to noise ratio of the recorded spectrum decreases. We present the results
of theoretical spectral simulations for formaldehyde (HCHO) that show how path length, interfering water concentration, and HCHO target concentration affect the expected differential absorption spectrum of the HCHO target at 1 ppb. HCHO serves as a proxy for any low abundance trace gas with relatively weak absorption features, which makes it sensitive to changes in the performance and setup of the OP-FTIR system, as explored in this study. We demonstrate that optical path lengths > ~300 m are necessary for robust HCHO spectral signatures (at our typical random plus systematic noise levels). Next, we present
the results of two field experiments where the retroreflector array area was increased to collect a larger fraction of returning radiation, at two-way path lengths ranging from 50 m to 1300 m. We confirm that the larger retroreflector array resulted in a slower decrease in the signal as a function of optical path length. Finally, we perform retrievals of HCHO concentrations from spectra collected at the same field site and path length in Halifax Harbour during 2018 and 2021, with a smaller and a larger retroreflector array, respectively. We demonstrate that retrievals based on larger retroreflector array spectra exhibit ~2x higher
precision (average standard deviation in hourly formaldehyde data bins over two days), even though systematic errors remain in the fitted spectra, due to water vapour. Where systematic fitting errors in interfering species (e.g., water) are significant, a longer path may not be optimal for a given target gas, leading instead to biased retrievals; moreover, at very long optical path lengths the signal-to-noise ratio decreases with increasing water vapour due to broadband mid-IR spectrum signal reduction in water-saturated regions. We discuss factors to consider in the choice of path length and retroreflector array size in open-
path FTIR spectroscopy, which must be made with care.

# 1 Introduction

## 1.1 OP-FTIR technique applications and implementation

Open-Path Fourier Transform InfraRed (OP-FTIR) spectroscopy is an established technique used to measure boundary layer trace gas concentrations, historically in fenceline monitoring of industrial emissions (e.g., Schill et al., 2022), near traffic emission sources (e.g., You et al., 2017), and in biomass burning events (e.g., Paton-Walsh et al., 2014). It has recently been used to monitor greenhouse gas emissions from agricultural fields (e.g., Flesch et al., 2016; Lin et al., 2019; Bai et al., 2019) and in urban settings (Byrne et al., 2020), also during the COVID-19 pandemic (You et al., 2021). Additionally, OP-FTIR has been applied in measurements of marine shipping port emissions (Wiacek et al., 2018), air-sea greenhouse gas exchange (Wiacek et al., 2021; Hellmich, 2022) and volcanic emissions (e.g., Pfeffer et al., 2018).

Most details of the system used in this study have been previously described (Wiacek et al., 2018) and here we only recount the general features of a monostatic OP-FTIR configuration, which consists of a spectrometer (containing an active infrared source, the interferometer, and an infrared detector), transfer optics, telescope, and a retroreflector array separated from the spectrometer by a few hundred meters (the optical path travelled by the light is double the separation). Beyond this separation, the returning signal diminishes and measurement quality is degraded (see Section 1.3). The source generates broadband mid-infrared light that is collimated and modulated in the spectrometer, then expanded by the telescope and sent toward the retroreflector array; the expanded beam passes through the atmosphere along the measurement path until it reaches the retroreflector, where it is reflected back towards the telescope and spectrometer for detection as a time-domain interferogram. In this configuration, unmodulated stray light from the thermally emitting atmosphere is easily rejected as a constant offset at the detector and the time-domain interferogram is subsequently Fourier-transformed into an atmospheric absorption spectrum.

Underlying atmospheric trace gas concentrations are derived by iteratively fitting a simulated spectrum to a measured spectrum until the residual is minimized; the details of our retrieval process have been described elsewhere (Wiacek et al., 2018, and references therein). The retrieval result is a path-average trace gas concentration, which means that measurements are representative of the full atmospheric path, as opposed to a point, while retaining high precision. This is an inherent advantage of the open-path system in ambient atmospheric measurements: it provides a more representative sampling of the broader measurement location and minimizes the effect of localized emissions.

## 1.2 Retroreflector array overview and modifications

Retroreflector arrays are constructed from multiple smaller cube-corner retroreflector elements, which can have various coatings depending on the wavelength of light the system utilizes. In the case of infrared light, the most commonly used coatings are gold and silver, due to their high reflectivity in the infrared spectral region (Bennett, 1965). Each cube-corner retroreflector reflects incoming light back toward the source in a slightly translated (up to ~6 cm) but parallel path. This is

accomplished by combining three flat mirrors orthogonally (Fig. 1, left), which results in the incoming beam being reflected twice to achieve a direction change of 180°, i.e., a retro-reflection (Fig. 1, right). Typically, commercially available arrays consist of either 30 or 60 cube-corner elements for either 30 cm or 60 cm array sizes.

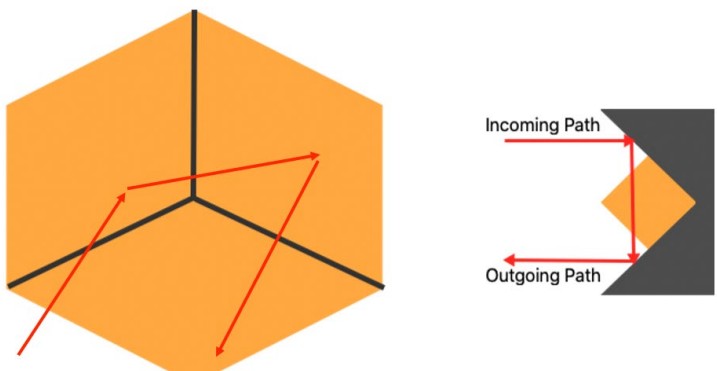

**Figure 1: (Left) single cube-corner retroreflector, front view. (Right) single cube-corner retroreflector, side view, with example incoming beam and two reflections shown by red arrows.**

To construct two larger than off-the-shelf retroreflector arrays for our work, 60 new cube-corner retroreflectors were purchased in 2020, each with a 63.5 mm outer diameter and nominal angle of acceptance of 30°. The beam deviation tolerance from parallel (after complete retro-reflection) is 20 arcsec (0.10 mrad). Each cube-corner retroreflector element was gold-coated (including a protective dielectric coating) to achieve 97% IR reflectivity. The pristine cube corners arrived at Saint Mary's University pre-mounted onto two custom array panels (30 new cubes on each panel, as required by the scientific investigation described in Wiacek et al., 2021) to which older cube corners (63 mm diameter, bare gold) were added from two existing 60 cm off-the-shelf retroreflector arrays. The smaller and older retroreflector array is shown in Fig. 2 (left), while the larger retroreflector array constructed using the older cube corners (degraded somewhat from ~180 days of cumulative field use since 2015) and the 30 pristine cube corners (representing a 50% increase in area) is shown in Fig. 2 (right). Due to the relatively high cost of pristine cube corners (~$300 USD each in 2020), the larger array could not be constructed out of entirely newly sourced cube corners. Note that the difference in cube substrate design resulted in a gap in close-packing the larger array (new cubes have a glass-substrate while old cubes have a metal substrate).

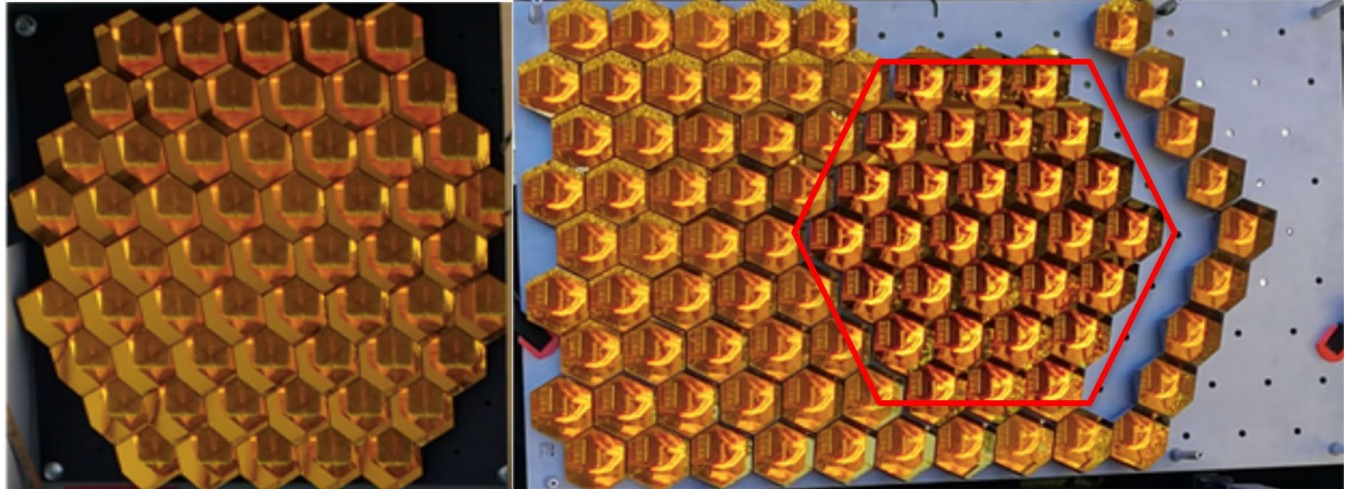

85  **Figure 2: (Left) pristine bare-gold retroreflector array (59 elements) acquired off-the-shelf in 2015. (Right) larger retroreflector array constructed in 2020 from 30 new and 59 used cube corners (new elements enclosed by red hexagon). The old and new cubes are not identical in substrate design and so could not be perfectly close-packed, resulting in a gap (right).**

### 1.3 Path length, beam divergence and retroreflector array overfilling

90  A critical variable in OP-FTIR spectroscopy is the optical path length (OPL), as it inversely correlates with the detection limit of the system for typically used paths; a longer path implies greater absorption and thus lower detection limits (Griffiths and de Haseth, 2007). At sufficiently long path lengths, however, beam divergence due to the necessarily imperfect beam collimation inside the spectrometer causes a decrease in returning signal due to overfilling of the distant retroreflector array (Fig. 3). In this simplified schematic of a mono-static arrangement, the co-located source (not shown), spectrometer (not

95  shown) and telescope (blue cylinder) are on the left, while the retroreflector array (60 cm or 120 cm) is on the right. The bi-directional red arrows show the IR radiation travelling along the open path, with a greatly exaggerated beam divergence. The larger array returns all – or at least *more* – divergent radiation from farther away towards the detector, depending on system and setup details, whereas the smaller array would be overfilled at that same distance. The spectrometer (3 mm aperture, 69 mm focal length) plus telescope (9:1 reducing) used in this experiment produces a 30 cm collimated beam, with an effective

100  beam divergence of ~1 mrad observed in the open path in field measurements, resulting in retroreflector array overfilling at a separation greater than ~150 m for our 60 cm retroreflector array. Finally, we note that array size cannot be increased *ad infinitum* in practice because cube-corner costs are prohibitive.

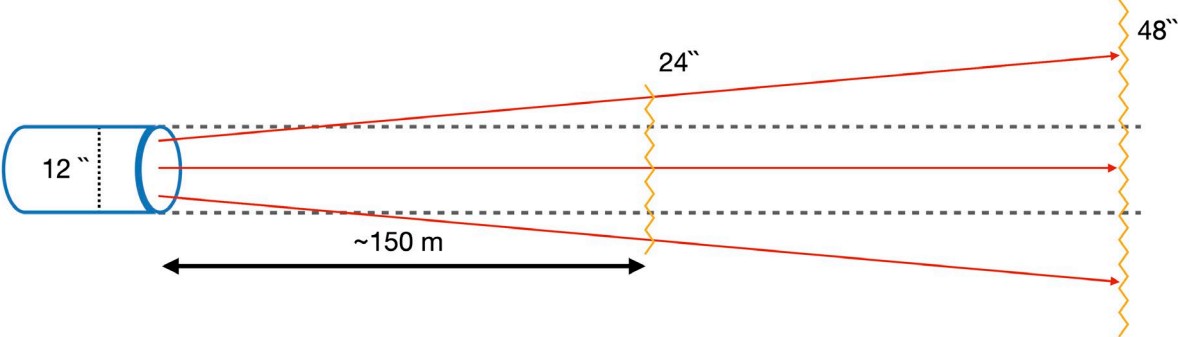

**Figure 3: OP-FTIR beam divergence (exaggerated) after leaving a 30 cm diameter telescope, leading to overfilling of a 60 cm retroreflector array in our system configuration (see text for details).**

## 1.4 Overview of study

The remainder of this paper is structured as follows. Section 2 outlines our experimental design, which first reviews the theoretical expectations for the absorption of formaldehyde (HCHO) in Section 2.1. Formaldehyde was chosen as the focus of our analysis because it is an important but challenging-to-measure ambient atmospheric constituent that is not reported by the ground-based National Air Pollutant Surveillance (NAPS) network in Canada. Nevertheless, it is detectable from space, e.g., by the TROPOMI instrument (October 2017 launch, 5.5 x 3.5 km$^2$ resolution after Aug 2019) and now by the geostationary TEMPO instrument (April 2023 launch, 2 x 4.75 km$^2$ resolution at centre of field of regard). TEMPO is expected to operationally produce hourly total columns of HCHO by October 2023. Further motivating the pursuit of more precise and diurnal surface HCHO products from OP-FTIR is the fact that surface HCHO concentrations have been derived from space-based total column measurements with the help of models (Zhu et al., 2017), but this is challenging work – both at the surface and from space (e.g., Vigouroux et al., 2020). Next, Section 2.2 describes two field experiments where path length was systematically increased and signal strength was analysed under conditions of different retroreflector array size. Section 2.3 discusses the setup of retrievals of formaldehyde concentration from field spectra recorded at the same very long path (and field site) but with different retroreflector array sizes (at different times). The technical design of retroreflector arrays, beam divergence and path length has been reported on before (e.g., Richardson & Griffiths, 2002, for FTIR; Merten et al., 2011, for UV), but to the best of our knowledge this is the first report of explicit tests of retroreflector array size in FTIR open-path measurements that vary path length, considering both theoretical expectations and retrieval results for a low abundance gas, the latter incorporating an analysis of systematic issues that arise in very long open-path experiments. Section 3 presents the results corresponding to Section 2, i.e., for spectral simulations, for increasing path in field experiments, and for increasing retroreflector array size in field experiments. Finally, Section 4 closes with a summary and key conclusions.

## 2 Experimental Design

### 2.1 Theoretical spectral simulations

The detection of trace gases by OP-FTIR depends on many variables besides the target gas concentration, such as the optical path length, interfering gas concentrations, environmental conditions (p, T, and IR beam extinguishing precipitation), and instrumental parameters. Simulations were conducted for the target gas formaldehyde to determine the theoretical minimum optical path length required for detection at 1 ppb, which is a background atmospheric concentration (Wiacek et al., 2018), although formaldehyde can reach 10 ppb in polluted environments (Wu et al., 2023). The path was increased from 50 to 1500 m while all other parameters listed in Table 1 were held constant, which resulted in stronger absorptions by both target and interfering gases. The increase in the target gas was isolated by calculating the differential absorption due to the target gas as the difference between a spectrum with the target and interfering gases, and another simulated spectrum with only the interfering gases (also using Table 1 input parameters, except setting the target concentration to 0 ppb). We performed these simulations using the forward model in the Multiple Atmospheric Layer Transmission (MALT) NLLS retrieval suite (Griffith, 1996; Griffith et al., 2012), which uses the HITRAN spectral database (Gordon et al., 2022) for line parameters; in this work we used HITRAN 2012 (Rothman et al., 2013).

**Table 1: List of input parameters used in spectral simulations.**

| | |
|---|---|
| Target Gas | formaldehyde (1 ppb, 0 ppb) |
| Spectral Window | 2700 cm$^{-1}$ - 2900 cm$^{-1}$ |
| Pressure | 1013.25 mb |
| Temperature | 15 °C |
| Optical Path Length | 50 m - 1500 m |
| Path Step Increase | 50 m |
| Interfering Gases | water (1%), methane (2 ppm), nitrous oxide (330 ppb) |

### 2.2 Impact of increasing path in field experiments on measured signal levels

To empirically determine the point at which effective beam divergence results in overfilling of a retroreflector for a particular system, an experiment can be conducted where open-path spectral measurements are made at successively increasing path lengths. This experiment also gives an indication of the overall effectiveness of the system at increasing path lengths for detecting a particular trace gas in a given spectral window while incorporating the practical effects of the increasing absorptions due to any interfering species (usually water vapour). In May 2015, such an experiment was conducted with a smaller retroreflector array constructed of, at the time, 59 pristine bare-gold cube-corner retroreflectors. The retroreflector array was progressively moved further from the spectrometer starting at a separation of 50 m and increasing to 450 m in 100-m

increments. This resulted in optical path lengths of 100-900 m. At each increment, spectroscopic measurements were made and the return signal level was recorded to assess the effects of beam divergence. Measurements took about 15-20 minutes per separation, including the time to relocate the array and re-align the telescope. In October 2020, this experiment was repeated, again over the course of a few hours, with the newly constructed larger (89 element) retroreflector array. We used spectrometer-array separations from 25 m to 650 m in approximately 100 m increments, with the goal of assessing the benefits of the larger

retroreflector array area, specifically with regard to overfilling.

At each location and path, we followed the manufacturer's recommended iterative procedure to focus the telescope on the retroreflector array (first a z-adjustment of the secondary mirror (in-out from primary mirror) followed by an x-y adjustment of the telescope (left-right, up-down), repeated at least once to maximize IR signal levels). The OP-FTIR system used was a

170 Bruker Open Path System, comprising an internal IR source, a modified 30 cm Schmidt-Cassegrain telescope, and a broadband IR detector responsive between 650 $cm^{-1}$ and 6500 $cm^{-1}$. Data acquisition was handled by the proprietary software OPUS RS, developed by Bruker, with the instrument operating at its maximum spectral resolution of 0.5 $cm^{-1}$ (Wiacek et al., 2018). Higher spectral resolutions are not typically used due to strong pressure-induced broadening effects in the near-surface horizontal path, of particular importance for large molecules whose rotational structure is not resolved (e.g., Hart and Griffiths,

2000, for classical least squares regression).

## 2.3 Impact of increased retroreflector array area on retrieved concentrations of HCHO

To further analyze the effect of the larger retroreflector array area on measurements, formaldehyde retrievals were performed on OP-FTIR absorption spectra recorded at the same location in the summer of 2018 (using the smaller retroreflector array) and the summer of 2021 (using the larger retroreflector array). These measurements were conducted across Halifax harbour,

with a spectrometer-array separation of ~560 m (Wiacek et al., 2021). As discussed above, the data acquisition of these measurements was conducted using the proprietary software, OPUS RS, at a resolution of 0.5 $cm^{-1}$, then apodized with the Norton-Beer 'medium' function (Norton and Beer, 1976; Norton and Beer, 1977). Trace gas retrievals were performed with the Multiple Atmospheric Layer Transmission forward model (Griffith, 1996) and a non-linear least squares iterative fitting routine. Spectra used as inputs to retrievals were filtered for IR intensity (at 2100 $cm^{-1}$) greater than 0.15 arbitrary units. These

filtering criteria exclude only poor weather conditions such as fog or heavy rain, which are the main cause of poor retrieval results; they do not exclude any long-path data that have lower returning signal levels due to divergence and retroreflector overfilling. HCHO spectral retrievals were performed between 2745 $cm^{-1}$ and 2800 $cm^{-1}$, with interfering species $H_2O$, $CH_4$ and $N_2O$, which all have significant absorptions in this window. Pressure and temperature data were obtained on an hourly basis from an Environment and Climate Change Canada (ECCC) meteorological station 7 km away (also coastal and at 5 m

above sea level). Other fitted parameters included the background transmission (a first order polynomial) and instrumental parameters of phase shift (affecting line symmetry) and spectral shift (affecting line position). Instrumental field of view and

effective apodization parameters were held constant, as in the work of Paton-Walsh et al. (2014) for HCHO. Retrieval results were filtered for root mean square (RMS) residual values less than 0.01 (1% mean fitting error).

# 3 Results

## 3.1 Theoretical spectral simulations

Fig. 4 shows the effect of increasing the optical path length (which starts at 50 m and increases up to 1500 m in 50-m increments) on the combined absorption spectrum of the target and interfering gases. The peak absorption is ~55%, corresponding to water features at approximately 2720 cm$^{-1}$. Absorption increases with increasing path length and so at shorter optical path lengths (< ~300 m) the water features reach only ~15% absorption. This means that detection limits are theoretically better at longer path lengths (if noise is constant), with the precise minimum optical path length different for each individual target gas (in the absence of any interfering species). Yet the effect of interfering gases cannot be ignored in practice and the simulated differential absorption of only the target gas is next shown in Fig. 5.

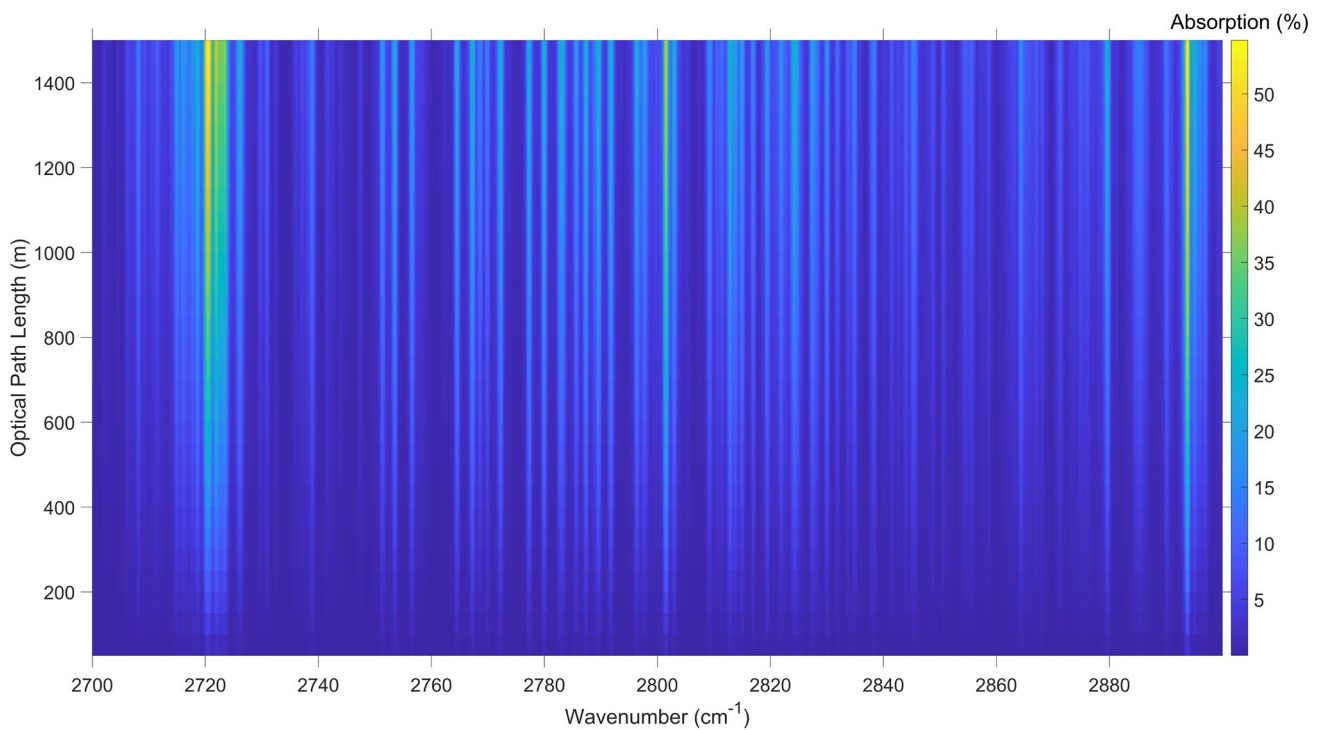

**Figure 4: Percent absorption due to 1 ppb formaldehyde and interfering species (see Table 1) as a function of varying optical path length and wavenumber.**

In Fig. 5 (top panel), the transmittance spectrum is cropped to better show the spectral features of formaldehyde, which only reach a minimum transmittance of 0.9984 at maximum path difference. The peak differential absorption formaldehyde features (brightest pink bands in bottom panel) occur mainly between 2760 cm$^{-1}$ to 2840 cm$^{-1}$. We see that short optical path lengths (< ~300 m) provide an approximate (simulated) differential percent absorption due to HCHO alone of < 0.04%, which is nevertheless technically large enough for formaldehyde to be detected in the presence of interfering gases because the random spectral noise of the measurements (Fig. 7, bottom panel) is ~2.6 lower than this value at these separations (i.e., ~0.015%, SNR ~6700). The differential percent absorption of formaldehyde alone reaches a maximum of < 0.14% at the maximum path length of 1500 m, which is only a 3.5x increase for a 5x path increase due to nonlinear transmittance effects caused by the overlap with water; this is still ~6x higher than the measurement random noise at these separations (i.e., ~0.023%, SNR ~4300). This means that at longer path lengths the interfering gases are not obscuring the target gas, i.e., the pink differential absorption bands intensify toward the maximum simulated path difference. However, it must be noted that this analysis considers random measurement noise only, and that systematic noise due to, e.g., spectroscopic parameter errors is always an issue in spectral fit residuals. For robust detection to be possible, the individual yet spectrally correlated features from the target gas should at least be comparable to the RMS noise levels of the retrieval (in the selected analysis window), which include both random and systematic components.

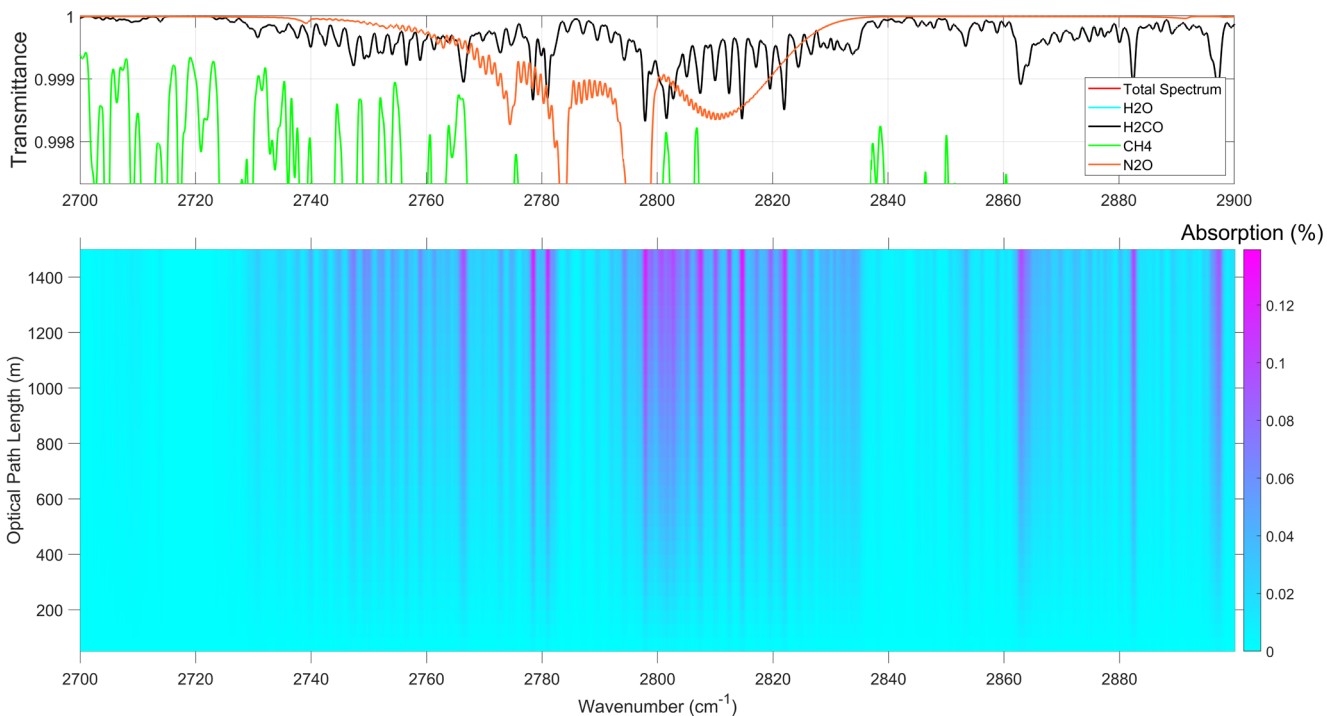

**Figure 5: (Bottom panel)** differential absorption due to 1 ppb formaldehyde in the presence of interfering gases (see Table 1) as a function of varying optical path length and wavenumber. **(Top panel)** corresponding individual transmittance spectra for target and

interfering gases (at the maximum optical path length in the bottom panel); the transmittance of water and thus the total spectrum is everywhere lower than the y-axis limits and on average 0.74 between 2760 – 2840 cm$^{-1}$.

## 3.2 Impact of increasing path in field experiments on measured signal levels

In the 2015 experiment ("Franklyn Street", smaller retro), it was determined that at approximately 150 m of separation between the spectrometer and retroreflector array (OPL of 300 m), the retroreflector became overfilled and the IR signal intensity (diagnosed at 2100 cm$^{-1}$) was progressively reduced at increasing paths up to a maximum tested separation of 450 m (OPL of 900 m). Fig. 6 shows the maximum signal level at each separation (at 2100 cm$^{-1}$) normalized by the value at the shortest OPL in each experiment. In the 2020 experiment ("Otter Lake", larger retro), the signal also began decreasing, more slowly with optical path length, somewhere between 300 m and 500 m, corresponding to theoretical paths where overfilling should begin for the shorter and the longer dimensions of the rectangular array, respectively. We fitted the theoretical form, for a circular geometry, of the ratio of a constant beam area to the expanding beam area as a function of the separation (*OPL/2*), using only those data points where overfilling has begun (expanding beam area is now greater than the size of the retroreflector array and the ratio is less than one). In this formulation, the radius of the constant beam area, $R$, corresponds to the radius of the retroreflector array (perfectly filled), the half-angle beam divergence is $\theta/2$, while $s$ is the separation shift required in each data set such that the ratio is 1 just as the array becomes perfectly filled. Franklyn experiment data were shifted by $s = 150$ m during fitting, which assumes perfect filling at 300 m, while Otter Lake data were shifted by $s = 200$ m during fitting, which assumes (imperfect) filling at 400 m (a compromise value due to the rectangular retroreflector geometry).

$$Ratio\ of\ signal\ decrease\ (\frac{OPL}{2} - s) = \frac{Constant\ beam\ area}{Expanding\ beam\ area\ (\frac{OPL}{2} - s)} = \frac{\pi \cdot R^2}{\pi(R + (\frac{OPL}{2} - s) \cdot \frac{\theta}{2})^2}$$

To be able to plot underfilled and overfilled normalized intensity data together in Fig. 6, the fitted lines are plotted without the separation shift and as a function of OPL instead of separation (*OPL/2*). Since the Otter Lake retroreflector is rectangular (unlike the assumed mathematical form), the signal level at 500 m is lower than the theoretical curve predicts, but the discrepancy is reduced for greater optical paths. This greatest misfit at 500 m is consistent with the fact that the percentage of the beam area lost due to a rectangular retroreflector array, as compared to the percent area lost simply due to the circular beam spreading, is maximized at the true point of overfilling. Overall, the data agrees with theory remarkably well in functional form, especially given the coarse sampling as a function of OPL in the region of transition from underfilling to overfilling.

It will become relevant in the following noise discussion to note that the absolute maximum signal levels underlying Fig. 6 are 25% lower at Otter Lake (~0.69, 100 m) as compared to Franklyn Street (~0.92, 100 m) despite the 50% larger retro array. The cause for this decrease in signal level is likely a combination of factors, including the 2x higher number of pristine cubes

in 2015 and a decrease in the combined Globar$^{TM}$ infrared source intensity and detector efficiency by ~10% between 2015 and 2020, together with the unknown accuracy of orthogonal retroreflector array alignment with respect to the IR beam (major signal factors like detector pre-amplifier gain and spectrometer aperture were the same between experiments).

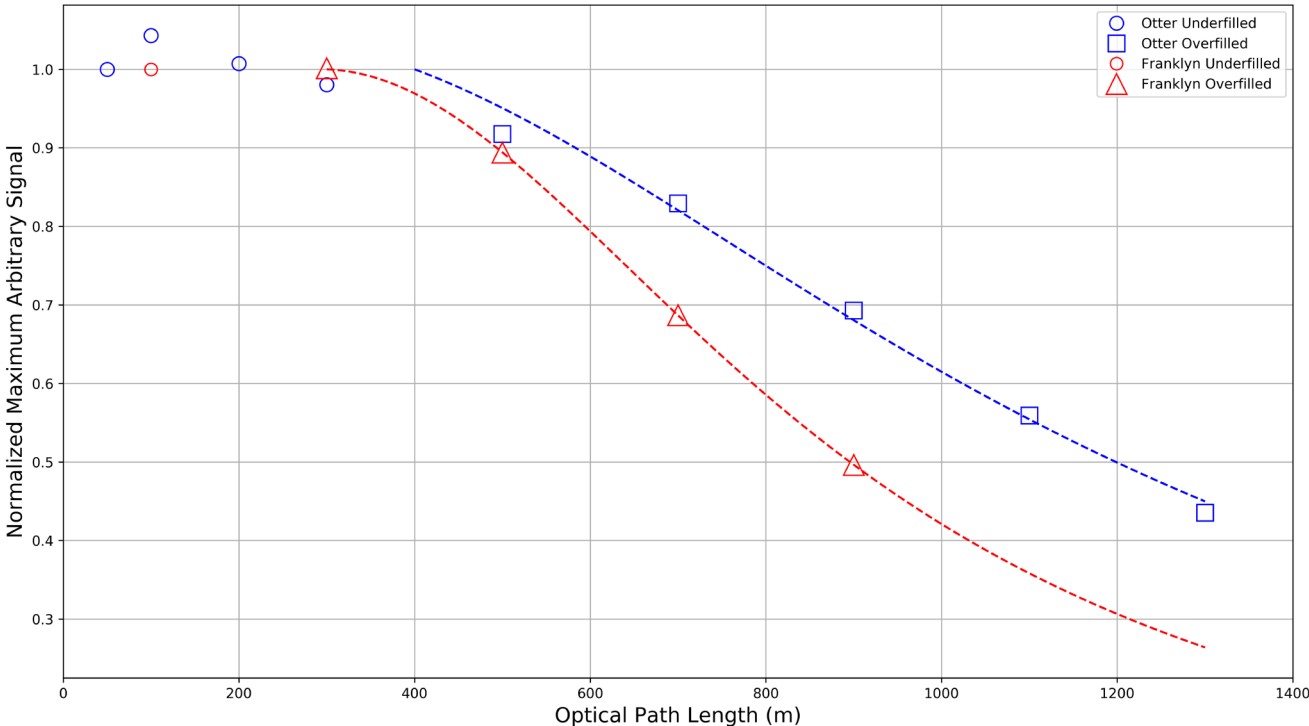

**Figure 6: Normalized maximum arbitrary IR signal intensity at 2100 cm$^{-1}$ for varying optical path lengths at Franklyn Street (May 27, 2015, 59 cube-corner retroreflector) and Otter Lake (October 16, 2020, 89 cube-corner retroreflector), together with the fitted curves for decreasing signal levels (see text for details).**

To determine the instrumental noise in each experiment, the standard deviation was calculated after de-trending the raw spectra in the region of 7640 cm$^{-1}$ to 7740 cm$^{-1}$ (outside of the detector's range of response where only instrumental noise is present) by fitting a 3$^{rd}$ order polynomial function to the region. The de-trending removed any change in the mean signal, thus making the signal in the region vary about zero so that a true random noise value could be calculated. The detrended signal in the spectral region used to estimate instrumental noise is shown in Fig. 7 (top) for the Otter Lake experiment. It has random noise characteristics and there are no correlated spectral absorption features in this spectral region between spectra with different optical path lengths (as expected). The same method of noise calculation was also applied to the Franklyn Street data in the same region (not shown). The signal-to-noise ratio (SNR) for both experiments was calculated using the maximum spectral signal at each path length and dividing by the instrument noise. As expected, the longer paths, where the retroreflector is overfilled and the returning signal is being lost, exhibit lower SNR values at both locations (Fig. 7 bottom), however, this is

complicated by the fact that the noise itself is not constant (see legend of Fig. 7, top, for Otter Lake, similar for Franklyn Street, also not shown). One must also be cautious not to over-interpret the SNR values between the experiments because there are absolute signal differences in addition to the changing noise, as discussed previously. Nevertheless, there is some indication that SNR is decreasing more slowly with optical path in the larger retroreflector array experiment.

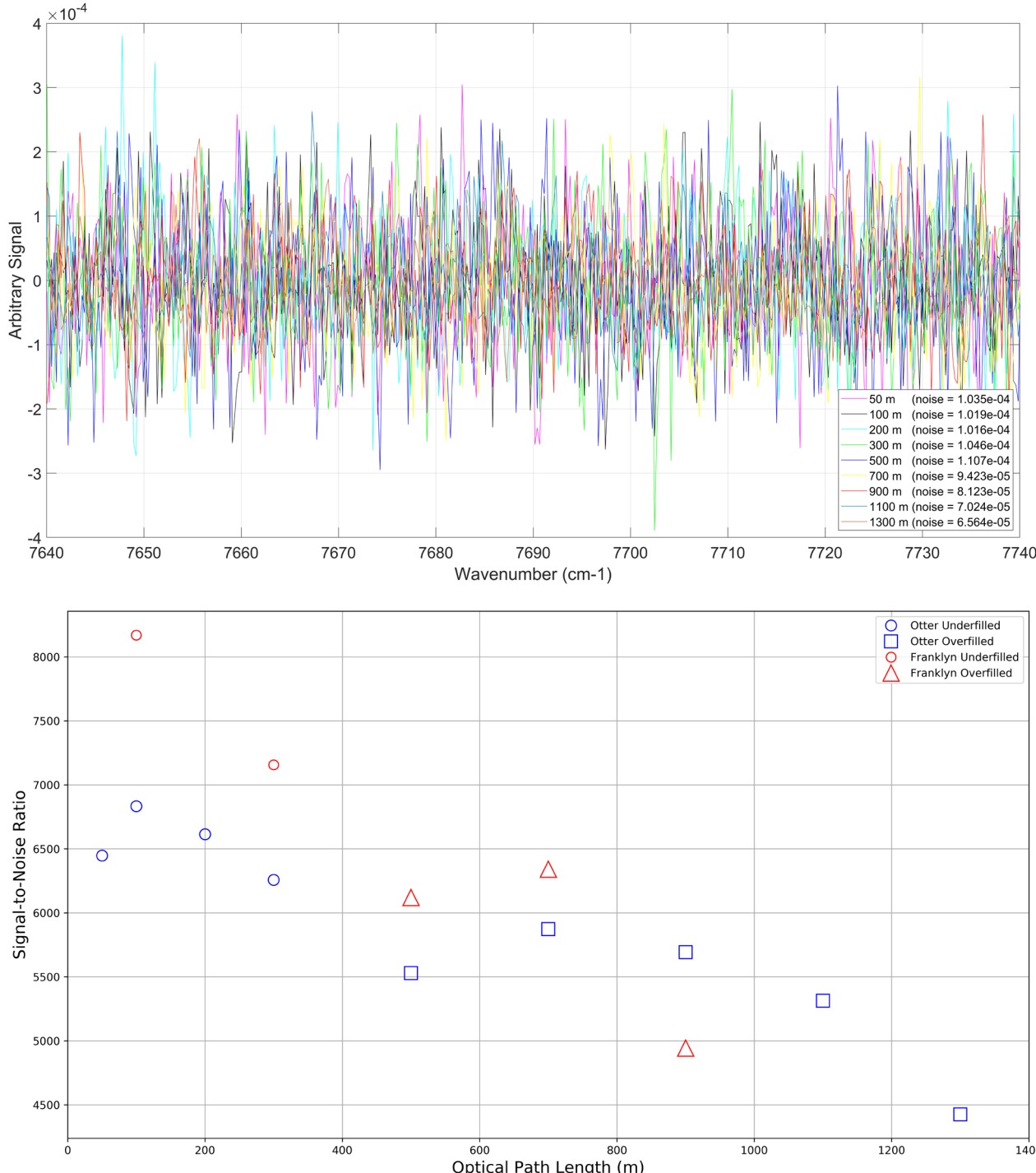

**Figure 7: (Top)** De-trended spectral signatures in a region outside of detector response at Otter Lake, which correspond to noise values. The standard deviation of the de-trended signal, i.e., noise, at each optical path is given in brackets. **(Bottom)** SNR values at Franklyn Street and Otter Lake.

**3.3 Impact of increased retroreflector array area on retrieved concentrations of HCHO**

Before analysing the impact of the larger retroreflector array used in 2021 on HCHO retrievals, the impact of different acquisition durations must also be accounted for. This is because increased precision in retrieved concentrations is influenced by both increased *signal* levels due to a larger retroreflector array, and decreased *noise* levels due to a longer acquisition time, *t*. The acquisition time was 1 minute in 2018 and 2 minutes in 2021, therefore, 2021 data will have an increased SNR by a factor of $\sqrt{2}$ on account of the doubled acquisition time alone, since $SNR \propto t^{1/2}$ (Griffiths and de Haseth, 2007). We account for this effect and thus isolate the effect of the larger retroreflector array alone on the SNR of the measured spectra, using data from a two-month period of each dataset, starting in March of 2021 and August of 2018. The sampling geometry for each of these experiments was identical, with the larger retroreflector in 2021 being the only notable difference. These months represent the period with least data gaps and when the infrared intensity was at its peak for each field campaign, before retroreflector wear and tear increased. In Fig. 8, the SNR, calculated using the same method as in Fig. 7, is plotted for the two-month period in 2018 and in 2021. These are daily mean SNR values, filtered to remove measurements with low IR intensity due to poor weather. Additionally, the SNR values from 2021, adjusted for the longer acquisition time (reduced by a factor of $\sqrt{2}$), are also plotted. The mean SNR value in 2021 is ~3x higher than the mean SNR value in 2018 ($\sim 5400$ vs. $\sim 1950$), while the adjusted mean SNR for 2021 is ~2x the mean SNR of 2018 ($\sim 3800$ vs. $\sim 1950$). This shows that even when accounting for the longer acquisition times in 2021 (by reducing the SNR), the quality of the spectral measurements is still doubled by the larger retroreflectors. Given a 50% increase in retroreflector area, we expected a 1.5 factor of increase based on area, instead of 2. Two reasons may be responsible for the discrepancy. First, the signal levels are sensitive to the orthogonal alignment of the array to the beam, and this alignment was more carefully considered in 2021 than in 2018, possibly giving higher 2021 signal levels. Second, the signal level in 2018 (August) is also appreciably lowered as compared to 2021 (March) because of increased water vapour levels in August (see the effect of water vapour on SNR in broadband FTIR spectra below and in Fig. 12).

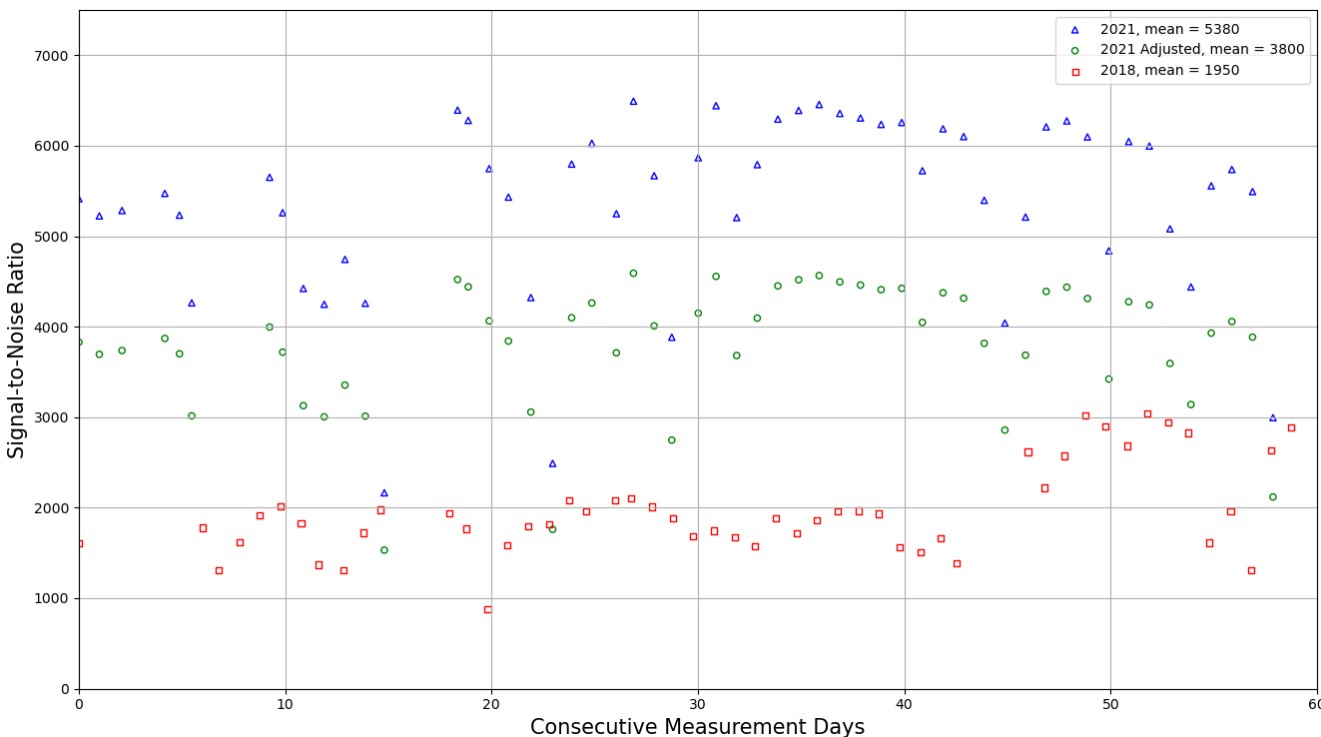

Figure 8: Daily mean signal-to-noise ratio in two-months of spectra in 2021 (large array), 2018 (small array), and "adjusted 2021" (see text for details).

Now we discuss the practical effect of a larger retroreflector array on the retrievals of formaldehyde from absorption spectra recorded with the same sampling geometry. The field experiments in 2018 (smaller array) and in 2021 (larger array) were

315 both conducted over a 1120 m optical path length, however, the data was gathered during largely non-overlapping seasons. To determine a reasonable comparison timeframe between 2018 and 2021, we looked for conditions of similar specific humidity due to the pervasive effects of water vapour in infrared spectra and on retrieved HCHO, particularly at long paths. Specific humidity was calculated using the relative humidity, temperature, and pressure (sourced from the ECCC station , as discussed previously). Additionally, we were interested in time periods with similar temperatures and thus the two-week period from the

320 1st to the 14th of September in 2018 and the 1st to the 14th June in 2021 was selected. A summary of the key acquisition data for each field experiment is given in Table 2.

**Table 2: Summary of key acquisition data for each field experiment in Section 3.3.**

| Year | Retroreflector Dimensions [cm] | OPL [m] | Acquisition Time |
|---|---|---|---|
| 2018 | 60 x 60 | 1120 | 1 minute |
| 2021 | 60 x 90 | 1120 | 2 minute |

| 2023 | 60 x 60 | 440 | 4 minute |

Fig. 9 shows the formaldehyde concentration retrieved for September 2018 and June 2021, with colour showing the infrared intensity at 2100 cm$^{-1}$. In 2021 both the acquisition time was longer (2 vs. 1 minutes or 480 vs. 240 co-adds at 4 Hz) and the sampling strategy was different in that observations were made by alternatively pointing to two different retroreflector arrays. This means that 2021 observations yield four times fewer spectra in each hour, which can be seen clearly in Fig. 9. Infrared intensity is, on average, higher in 2021 than in 2018 by ~50% (~0.69 vs. ~ 0.44) due to the larger retroreflector resulting in greater returning radiation, and a higher SNR. Correspondingly, the scatter (precision) of the 2021 formaldehyde retrievals is smaller (better) as compared to the scatter of the 2018 retrievals, which is quantified next in Fig. 10.

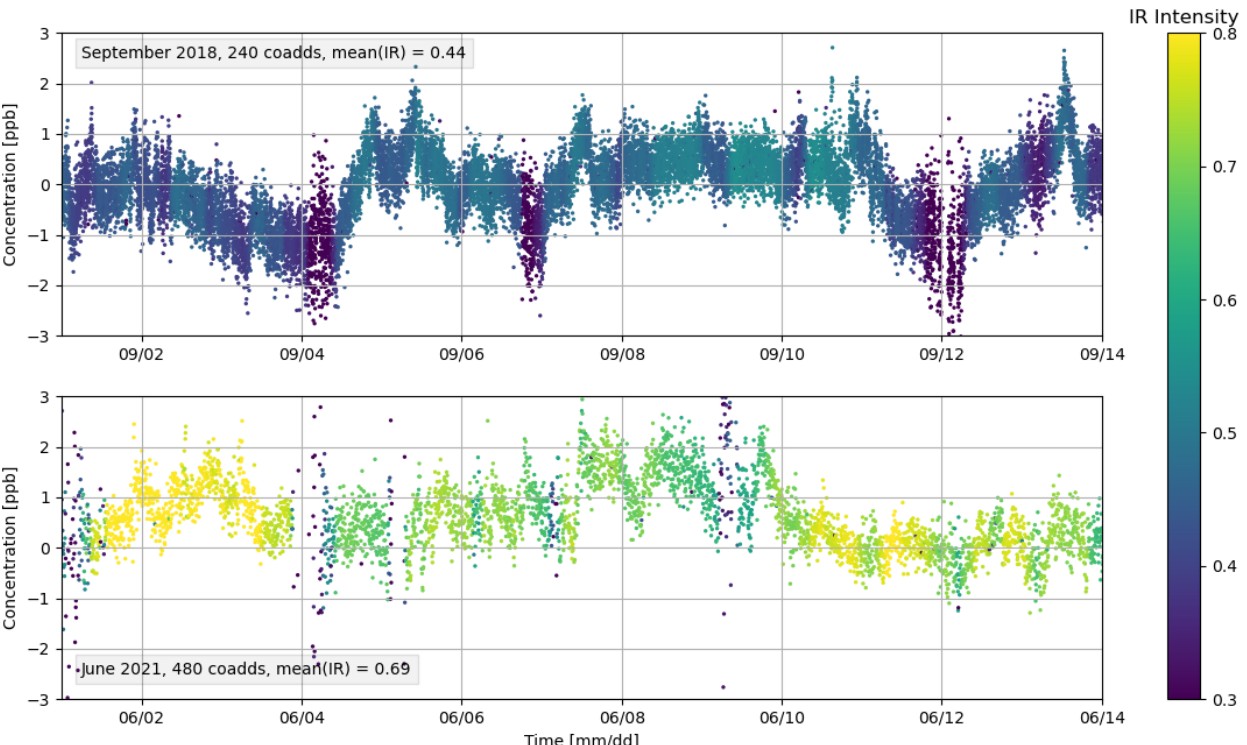

**Figure 9: Retrieved formaldehyde [ppb] for (top) September 2018 (small array) and (bottom) June 2021 (large array), where colour represents spectral infrared intensity in both time periods. 2018 measurements used the same field configuration but smaller retroreflectors than 2021 measurements. The acquisition length and sampling rate were different in each year (see text for details).**

Fig. 10 shows hourly mean retrieved HCHO concentrations (as boxplots) on September 12-13 in 2018 and on June 12-13 in 2021, chosen as days free from obvious ship emissions in our open path, which is traversed by ships. An hourly time frame was selected to minimize the impact of physical processes, such as daily photochemical variations or small and nearby HCHO

emissions, on the spread of the hourly means, i.e., so that it corresponds more closely to the true measurement precision. The 1st and 3rd quartiles of the 2021 data are closer to each other when compared to the 2018 data (average interquartile range of 0.47 ppb for 2021 and 1.05 ppb for 2018), and the average hourly standard deviation over the two-day span is smaller than in 2018 (0.35 ppb versus 0.68 ppb), indicating a better precision of the 2021 dataset. In part due to the lower precision, the diurnal pattern in 2018 is less discernible, while the higher precision 2021 data more clearly show the expected concentration minima during the night (0000 h - 0700 h) and maxima during the day (1000 h – 1800 h). (A rigorous examination of the strength of the diurnal signal in both datasets over all available days is outside the scope of this study.) Lastly, 2021 data exhibit fewer negative retrieved HCHO concentrations over a two-day period, which we discuss next.

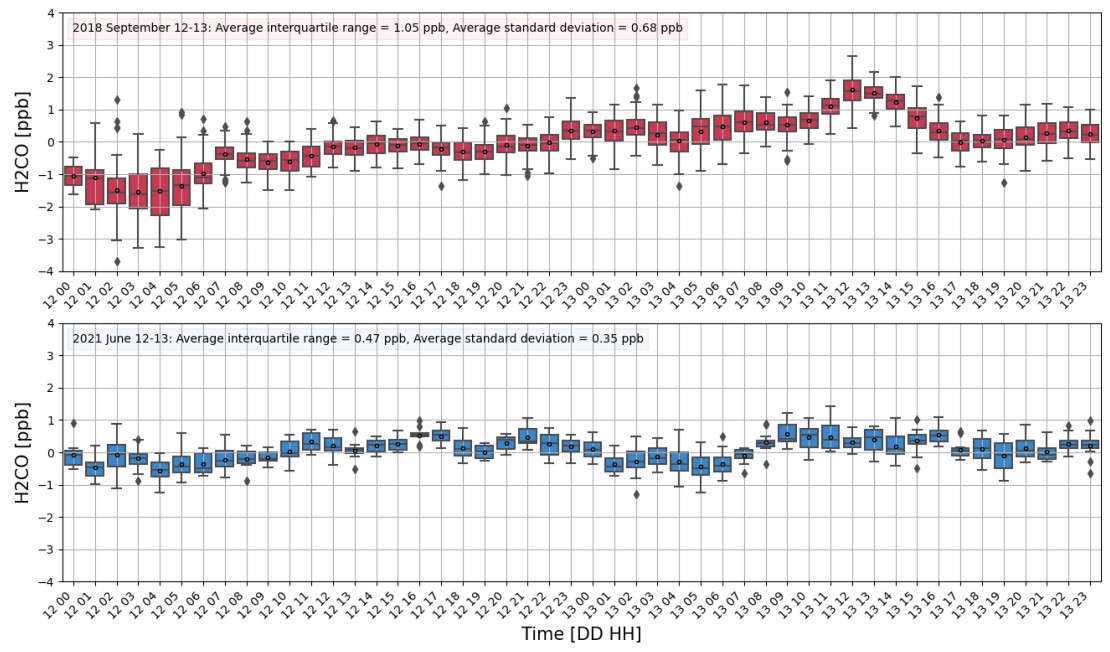

**Figure 10: Formaldehyde hourly mean concentrations [ppb] from (top) September 12 - 13, 2018 (small array) and (bottom) June 12 – 13, 2021 (large array). Horizontal lines represent median concentrations, circles represent mean concentrations, upper and lower box edges represent 75th and 25th concentration percentiles ("q3" and "q1", respectively), upper and lower whiskers represent maximum and minimum concentration limits (defined as +/-2.7σ (approx. 99.3 percentile)), and grey diamonds are concentration outliers.**

### 3.3.1 Impact of H2O on retrieved concentrations of HCHO

As discussed in previous sections (1.3), longer optical path lengths allow for greater absorption of trace gases. This results in a lower detection limit for a low-abundance target gas such as formaldehyde, however, this is also coupled with an increase in the absorption of any interfering gases, such as water vapour. Greater absorption by water has an impact on the retrieval process

via both systematic and random errors, which is something we noticed in analysing data from 2023 (not discussed up to now) that contained far fewer negative formaldehyde concentrations despite a lower spectral SNR and shorter optical path (lower formaldehyde absorption signal). First, a high absorption due to water increases the systematic misfitting to the measured spectrum. Fig. 11 shows the average $H_2O$ transmittance (top), spectral residuals as boxplots (middle), and spectral residuals

coloured by $H_2O$ transmittance (bottom) for a two-week period in June of 2021 and again in June of 2023; for clarity, only half of the retrieval window is shown (2772.8 – 2800.3 cm$^{-1}$). The measurements in 2023 were recorded over a shorter optical path length of 440 m *vs.* 1120 m in 2021 and 2018. (This short OPL is consistent with the minimum OPL for ~2.6x noise level detection of formaldehyde that we identified in simulations in Section 3.1.) The $H_2O$ transmittance in 2021 and 2023 (top panel) shows a similar structure as expected, however, the transmittance depth in 2021 is consistently double the transmittance

depth in 2023, overwhelmingly due to water vapour absorption features. The average spectral residuals (as boxplots for a two-week period, middle panel) show the increase in systematic misfitting at the longer path. The average of the (absolute value) median residuals at each point in the spectral window for the shorter 2023 path is 5x smaller than that of the longer 2021 path (see legend), i.e., the spectral residual for the shorter path varies less about zero and is lower overall. Additionally, the average interquartile range of the short-path dataset is half that of the long-path dataset (see legend), meaning that there is less variance

in the middle half of the short-path dataset residuals.

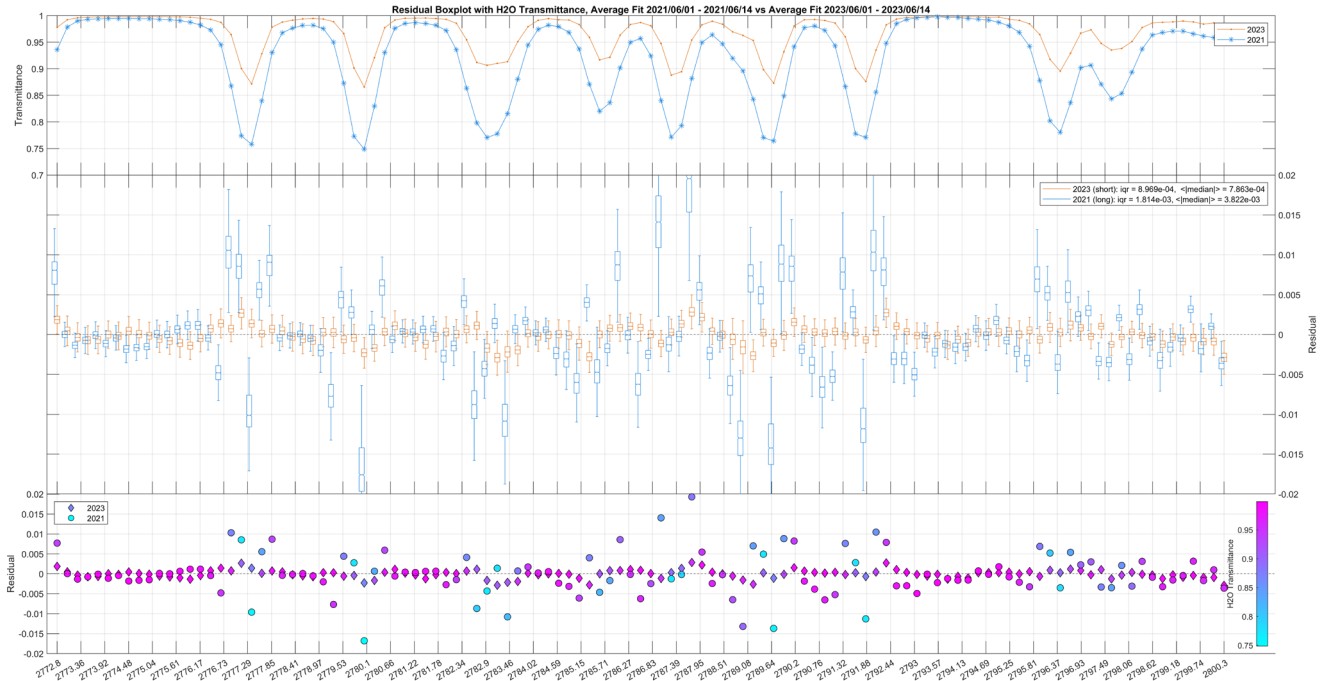

**Figure 11: All plot panels are based on data from June 1 - 14, 2021 (large array) and June 1 - 14, 2023 (small array) (Top) Average H2O transmittance spectrum for 2021 (blue) and 2023 (orange). (Middle) Spectral residuals from 2021 (blue) and 2023 (orange).**

**Boxplot features as in Figure 10. The legend gives the mean value of the (absolute value) medians and the mean interquartile range**

**in the spectral region shown. (Bottom) Average spectral residuals for 2021 (circles) and 2023 (diamonds), with colour indicating H2O transmittance.**

In the case of a low absorption gas like formaldehyde, a high abundance of water can result in a retrieved concentration that is negative, which is particularly pronounced during the summer months. Fig. 12 shows the retrieved formaldehyde concentration as a function of retrieved water vapour concentration (as boxplots) for 2021 and 2023, binned into 0.25% H2O increments. In both datasets the majority of measurements have a retrieved water concentration of 1.0% or less (~70% in 2021 and ~80 % in 2023); in these retrievals the median and mean retrieved formaldehyde concentration is positive. When the retrieved water concentration exceeds 1.0%, the 2021 median and mean retrieved formaldehyde concentrations become negative, while in 2023 they remain positive. The mean SNR in the 2021 dataset is higher than the corresponding mean SNR in the 2023 dataset for each box because in 2021 the retroreflector array was both bigger and comprised cubes in better condition. Nevertheless, ~40% of the formaldehyde concentrations retrieved in the 2021 dataset are negative, while less than ~2% are negative in the 2023 dataset, implying a formaldehyde concentration bias reduction. This is very likely a result of the lower transmittance due to $H_2O$ in the 2021 dataset compared to the 2023 dataset (Fig. 11, top) and the associated higher systematic misfitting of these water features (Fig. 11, middle) as the well-known spectroscopic database uncertainties are amplified by the long path. Lastly, we discuss how water impacts random errors. At the longer path used in this study (2021, blue, bold text insert in Fig. 12) the broadband spectral effect of water vapour is large enough to systematically reduce the SNR as water vapour abundance increases, whereas at the shorter path (2023, orange, bold text insert in Fig. 12) SNR is largely independent of water vapour. This is another mechanism by which pathlength reduces SNR, thereby increasing random errors, in addition to by overfilling of the retroreflector array.

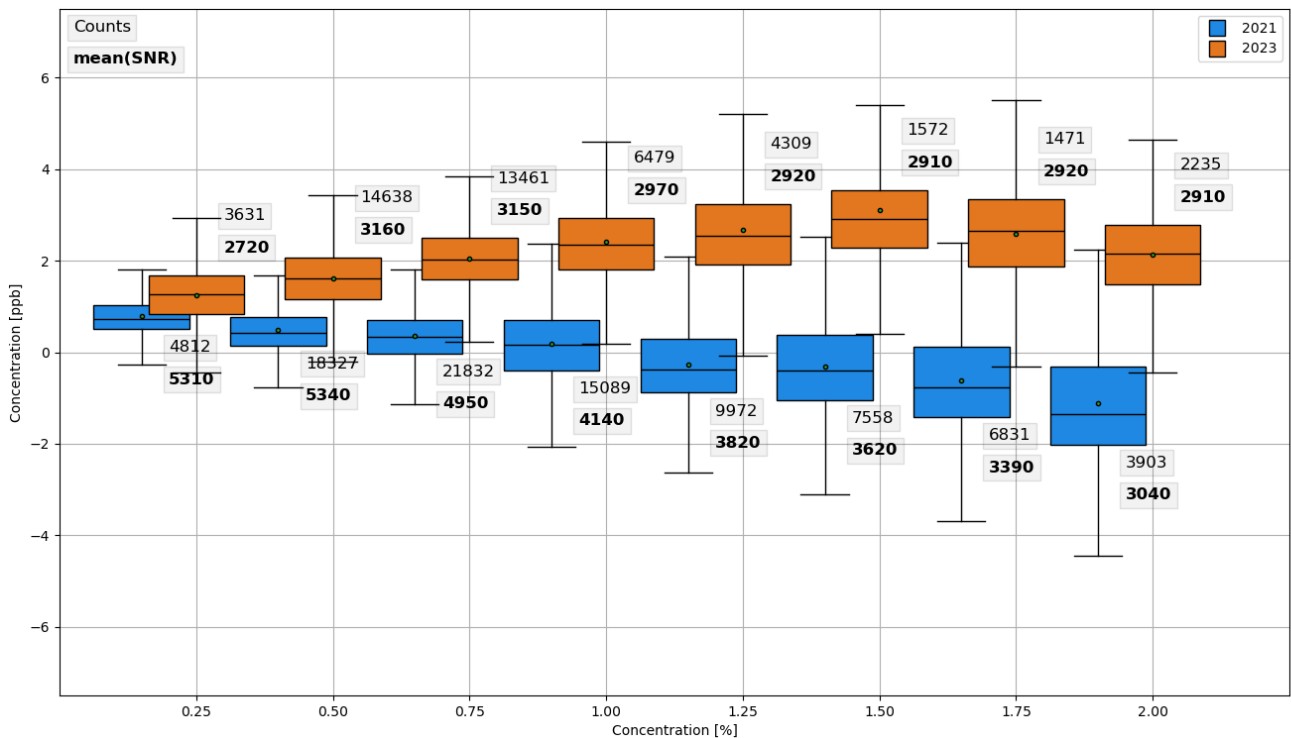

**Figure 12: Retrieved formaldehyde [ppb] plotted as a function of retrieved water vapour [%] from 2021 (blue, large array) and 2023 (orange, small array), with 2021 data offset for clarity. Text inserts give the samples per box on the top and the mean SNR per box on the bottom (in bold). Boxplot features as in Figure 10.**

## 4 Summary and Conclusions

Open-Path Fourier Transform InfraRed (OP-FTIR) spectroscopy is an established technique used to measure boundary layer trace gas concentrations. Diurnal measurements are possible at high temporal frequency, making OP-FTIR data suitable for the validation of space-based measurements of, e.g., daytime formaldehyde by the TEMPO geosynchronous instrument or day and night ammonia by the CrIS sun-synchronous instrument. A critical variable in OP-FTIR spectroscopy is the optical path length, as it inversely correlates with the detection limit of the system for moderate OPL values, where a longer OPL implies greater absorption and thus lower detection limits. At sufficiently long optical path lengths, however, beam divergence due to the necessarily imperfect beam collimation inside the spectrometer causes a decrease in returning signal due to overfilling of the distant retroreflector array. To the best of our knowledge, this is the first report of explicit tests of retroreflector array size in FTIR open-path measurements that vary path length, considering both theoretical expectations and retrieval results for a low abundance gas, the latter incorporating an analysis of systematic issues that arise in very long open-path experiments.

We performed spectral simulations for formaldehyde wherein the OPL was increased from 50 m to 1500 m while all other simulations parameters remained constant. Increased path resulted in stronger absorptions by both target and interfering gases,

as expected, with the differential absorption due to the target gas maximized away from its interfering species between 2760 cm$^{-1}$ and 2840 cm$^{-1}$. Based on consideration of absorption strength vs random error levels alone (spectral SNR) short optical path lengths (< ~300 m) result in an approximate differential percent absorption due to HCHO alone of < 0.04%, which allows formaldehyde detection at ~2.6x the noise level in our system. At an optical path of 1500 m the differential formaldehyde signal reaches < 0.14%, which is only a 3.5x increase for a 5x path increase due to nonlinear transmittance effects caused by the overlap with water; this is still ~6x higher than our system's random noise at these separations, however, at such long path lengths systematic errors in water vapour spectroscopy play an important role in formaldehyde retrieval results.

In 2015 and in 2020, we made OP-FTIR field measurements at progressively increasing path lengths using a smaller retroreflector array and a 50% larger retroreflector array, respectively. From these field experiments, we confirmed that the maximum spectral signal in the two experiments decreases with spectrometer-array separation as expected from theoretical considerations, with the maximum signal of the larger retroreflector experiment decreasing more slowly. While instrumental noise is not constant, there is some indication that SNR is decreasing more slowly with optical path length in the larger retroreflector array experiment.

In 2018 and in 2021, we made OP-FTIR field measurements at the same long path and location, but with a smaller and a 50% larger retroreflector array, respectively. After accounting for the longer spectral acquisition time in 2021, we found the mean daily average SNR in a two-month period corresponding to the highest IR intensity for both experiments to be greater by a factor of ~2x in 2021 data using the larger retroreflector array.

Next, we compared retrieved formaldehyde concentrations from the same measurement configuration during the first two weeks of September 2018 and the first two weeks of June 2021 based on similar environmental conditions (absolute water vapour and temperature). Given the higher SNR in 2021, the scatter (precision) of the 2021 formaldehyde retrievals is smaller (better) as compared to the scatter of the 2018 retrievals. In examining the hourly mean formaldehyde concentration statistics over two days, we find an average interquartile range of 0.47 ppb for 2021 and 1.05 ppb for 2018; the average hourly standard deviation over the two-day span was 0.35 ppb for 2021 and 0.68 for 2018 ppb. In part due to the higher precision, the expected diurnal formaldehyde pattern can be discerned clearly in 2021, but less so in 2018. Moreover, 2021 data exhibits fewer negative retrieved HCHO concentrations than 2018 data over a two-day period.

We also found that a 2023 short-path dataset with lower spectral SNR and lower formaldehyde absorption signal outperformed the 2021 dataset with far fewer negative retrieved formaldehyde concentrations (2% vs. 40%), indicating a formaldehyde concentration bias reduction. By analysing the spectral fit residuals we attribute this to spectral misfitting of strong water vapour absorptions in 2021, as the spectroscopic database uncertainties are amplified by the very long path conditions (a systematic error). Additionally, we document a second mechanism by which pathlength reduces SNR (associated with random

errors), in addition to via overfilling of the retroreflector array: at very long pathlengths SNR decreases with increasing water vapour conditions due to broadband mid-IR spectrum signal reduction effects in water-saturated regions. Work to optimize the negative retrieval bias is beyond the scope of this paper but ongoing and includes an examination of: updated spectroscopic database parameters for water (and other interferers) including their temperature and pressure dependencies; correlations between the target gas and other retrieved parameters (interfering species, instrumental parameters, continuum parameters); retrievals that also include temperature as a retrieval target; multi-step retrieval approaches where water can be retrieved first and then constrained in the HCHO window.

In summary, the choice of path length and retroreflector array size in open-path FTIR spectroscopy must be made with care. Longer paths increase target gas absorption (lowering detection limits) and larger retroreflector arrays improve the SNR of the spectra by increasing the return signal (improving retrieved concentration precision), but there are limitations to both. Open-path beam divergence of a given system determines the size of arrays needed to reduce overfilling at a given separation (driven by the detection limits of a particular target gas), but array size is constrained by prohibitive cost. Pathlength is constrained indirectly by 1) the accuracy of the spectroscopic parameters of water vapour, which may lead to systematic fitting errors and negative target gas concentrations, 2) the reduced SNR due to broadband water vapour absorption, and 3) the ability to point accurately enough at a distant retroreflector array. Nevertheless, there is an optimum array size and path combination to find in each specific observational environment and application, as explored in this work. In the case of HCHO, theoretical considerations dictate an OPL greater than ~300 m, while experimental results point to degrading retrieval quality via interfering species (primarily water) at our longest OPL of 1120 m in 2021, with no such degradation at an OPL of 440 m used in 2023.

*Data availability.* The OP-FTIR spectra underlying this study are in preparation for upload to the Canadian Dataverse Repository (https://borealisdata.ca/). Currently, spectra and concentration retrieval results are available upon request from the authors.

*Author contributions.* AW conceived of and designed the study, with input from CP. Both authors performed data acquisition and retrievals, examined and interpreted the results and prepared the manuscript.

*Competing interests.* The contact author has declared that none of the authors has any competing interests.

*Acknowledgements.* The authors thank members of the Wiacek Atmospheric Research Group for assistance with OP-FTIR field deployment in 2018 and 2021 (Taylor Gray, Morgan Mitchell, Ian Ashpole, Martin Hellmich) as well as Taylor Gray for developing the initial version of the differential optical absorption simulation codes.

*Financial support.* This research has been supported by the Canadian Space Agency (grant no. 16SUASMPTN), the Canadian National Sciences and Engineering Research Council through the Discovery Grants Programme (grant no. RGPIN-2014-03888, RGPIN-2022-05225), and Saint Mary's University.

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
