# Peer review of "On Path Length, Beam Divergence, and Retroreflector Array Size in Open-Path FTIR Spectroscopy"

_Atmospheric Measurement Techniques, 2024_

## Author Comment (AC1)

**Format Key:**

*Reviewer comments in blue italics with some cross-referencing **in pink.***
Author responses in black.

**RC1**: Anonymous Referee #1, 16 Aug 2024

*The manuscript describes three different approaches to analyze the dependency of open-path FTIR trace gas measurements on path length and retro-reflector array size. These approaches are: 1. simulation of absorption spectra for different path lengths, 2. dedicated measurements of spectral signal-to-noise-ratio (SNR) with the reflector array at different distances, and 3. analysis and comparison of two field deployments, which primarily differ in the size of the utilized reflector array. While this study focuses on the trace gas formaldehyde (HCHO), it aims also for more general results, especially where signal return and spectral SNR is discussed. The data presented here has the potential to provide important insights on how the choice of reflector array size and path length influence the performance of open-path FTIR instruments, and, further, might provide some results applicable to other open-path techniques. However, in my view, the manuscript in its current form contains several ambiguities, inaccuracies and also a few errors throughout the results, which in some cases impact the results and their interpretation significantly. Hence I recommend to publish this work in AMT, since it fits the scope of the journal perfectly, but only after major revisions, so it fits the journals technical standards. In the following, I address in depth my major points of critique and provide a detailed list of general remarks and line-by-line comments and technical corrections. I hope my thoughts and input on this matter is helpful for the authors moving forward.*

We thank RC1 for their thorough reading of our submission and their many thoughtful comments, which have greatly improved our manuscript.

***Major point**: Quantification of SNR dependency on path length (L. **20f**, L. **224f**, Figure 6, L. **247**, Figure 7)*
*I strongly disagree with the linear fits in Figure 6. Fitting a line to this data is neither supported by the data itself, nor by prior knowledge of the system. In a simplified model describing the overfilling of the reflector array you could model the diameter of your collimated beam by the telescope diameter $d_0$ plus the increase due to divergence $2 * \alpha * s$, where $\alpha$ is your divergence half angle and s your distance from the telescope. As long as this is smaller than your reflector diameter, all light should be returned. This is also what your data supports quite clearly for Franklyn Street and also somewhat for Otter Lake. The signal only decreases, once the diameter of the beam is equal to the diameter of the reflector array. According to your 1 mrad divergence, this should be the case for a distance of 150m, so right at 300m two-way path length. What happens after that is less clear. From our simplified model, we should assume a quadratic drop*

*of returned signal $(d_r / d_s)^2$ with $d_r$ being the reflector diameter and $d_s$ the beam diameter at distance s. Your data however seems to indicate a linear drop, so I could understand if you fit a line to it for empiric reasons, but in my opinion this should only be done for data points where overfilling is clearly happening. It would be interesting to get behind the reason for this linear behavior and where the simplified model from above breaks down, but I understand that this might not your focus or main interest for this publication. I would assume that some "fuzziness" in the transition region between the two regimes takes place, making the transition less harsh. This might be caused by the combination of turbulent seeing effects and the until here disregarded displacement of the reflector cubes. The Franklyn street results might just be a combination of this blurring in combination with only 3 data point in the region of overfilling. For Otter Lake, the model obviously breaks down, because the reflector array is not close to circular and the ratio of the beam which hits the reflector is not just $(d_r / d_s)^2$ as above. But the overall shape of the data still fits the idea, since a real decrease of signal only happens slightly later as for Franklyn street and the reflector array is slightly wider (in its shorter dimension) as for Frankly street. In a bit of a stretch, the data could also show an accelerated signal drop from roughly 700m onward, when the beam should now be large enough to overfill the reflector array in the larger dimension, but I would think that this is difficult to argue without better knowledge of the uncertainties of each data point.*

As noted by the reviewer, we indeed chose to fit the data presented in both figures with a straight line for empirical reasons (the data looks linear), while understanding that a $1/r^2$ dependence is expected in theory for a circular beam and retroreflector, and only *after* overfilling begins. The reviewer is also right in thinking that our main interest was not to explore the details of where the $1/r^2$ model breaks down, especially for the non-circular array. (This is why we explored the effect of larger arrays from three different angles, i.e., also using theoretical simulations of the benefit of longer optical paths in the presence of interfering species, and by examining the change in measurement precision in actual concentration retrievals based on field data.)

Nevertheless, these are very valid suggestions, and we have fitted the $1/r^2$ model to our data, only for separation points where overfilling is taking place and for normalized IR Intensity (Figure 6). We now also discuss departures from this model in the text, incorporating the comments above and making changes to the trend-related wording on (L224f).

***Further**, I think the absolute values in Figure 6, partially Figure 7, and especially for you slopes (if you stick to them) are not that helpful here. Such numbers are already highly dependent on the concrete setup (beam divergence, light throughput, reflector array) but if you would use relative numbers (so norm everything by your starting intensity or SNR and give your slopes in*

*relative intensity drop or relative SNR drop per distance) it would at least be independent of your concrete measurement mode, i.e. the measurement time, resolution, co-addition, etc.*

We have abandoned the linear slopes (previous comment) and we have switched to relative values instead of absolute values for Figure 6, since absolute signal levels are indeed highly system dependent.  We left absolute SNR values in Figure 7 since these can be compared between different systems, however, we abandoned any model fitting to SNR data (see next "Major point").

***Lastly*** *the data points in the lower panel of Figure 7 puzzle me, since they seem inconsistent with Figure 6: While the signal is the same for the first two data points for Franklyn Street (Fig. 6) the SNR differs by more than 10%. This means, that there is a sudden noise increase by more than 10% even though the signal stays the same. This should not be the case if your instrument works properly and is (somewhat) shot noise limited. But even if the noise is dominated by another (instrumental) noise source, this should not change between measurements. But if there is some external reason for this, the magnitude is on the same order as your trend (see the two data points), making it difficult to disentangle this effect from the trend you are interested in. At list I would ask for a proper treatment of this uncertainty in the plot and the fit. It is also not only these two data points. Throughout both datasets the trend is way less clear then for Fig. 6, indicating a large uncertainty as a result of the noise estimation.*

The reviewer highlights a subtle point that can be discerned between Figures 6 and 7, i.e., that noise values are non-constant and themselves 'noisy'.  We have abandoned model fitting in Figure 7 (bottom) and hence no longer report these trend values in the abstract (**L20f**) nor discuss them in the text.  We quantified the noise decrease with path in Figure 7 (top) and we modified the related discussion (**L247**) to acknowledge that the noise is not constant in each experiment. For that matter, there are absolute signal differences between the experiments too (already discussed in the paper and clarified via comment (L229), which makes comparisons between absolute SNR values less useful than comparisons of how the change with optical path.  While the type of noise dominating the measurements was not the objective of our study, this is discussed a bit more at (L269f), where the noise type is raised again by the reviewer.

***General remarks***
*Detection limit as a function of path length: The detection limit is only directly proportional to the path length if the SNR is independent of the path length, since the detection limit (how I think you define it here) is basically the point where the spectral response is on the order of the noise level. As you address yourself at several points within this manuscript, the **spectral response** (due to interfering species, and the exponential nature of absorption) and the **SNR** (due to a loss*

*of light) can both show a complex, non-linear dependency on path length. Since you like to analyse exactly the interaction of these two dependencies, I would try to avoid formulations which proclaim a direct dependency (such as directly proportional") between the detection limit and the path length.*

Good point. Reworded abstract from "The detection limit is directly proportional to the optical path length…" to "The detection limit at first increases with increasing path length…" (L10) The next part of this sentence (after semicolon) explains how beam divergence and array overfilling comes into play.

*Cube Corners shading themselves: Ad multiple points you mention, that large flat arrays of cube corners lead to increased shading between the cube corners. I don t think that this is true for the discussed situations, where the beam divergence stays the same (no modifications of the instrument or telescope) but the array has a larger distance to the instrument and, thus, the beam is larger in diameter. Even at this larger distance the maximum divergence of all the rays" within this beam stays the same and the cube corners are hit from the same angles independent of the distance to the instrument. Thus the shading should not change with distance, even with larger arrays (of course assuming the telescope points vertically at the center of the array).*

Agreed. We removed references to self-shading in Sect. 1.3 and Sect. 4.

*First report of retroreflector array size in open-path measurements: Considering FTIR open-path you might be right concerning the explicit testing, but I am not so sure about general open-path measurements. You might want to check out the DOAS open-path measurements in the UV and VIS spectral region, which go back at least to the 1990s, and where a lot of work on open-path optics was done. While I am no expert in this field and do not have an extensive overview, publications like Merten et al. (2011) addressed similar questions.*

We were indeed referring to FTIR and have made this clarification, along with including a few references to technical studies that explore array size and beam divergence, but not examine environmental data, like our study.

*Detection limit results in simulation: I can easily follow your description of your spectral simulations in subsections 2.1 and 3.1, but I do not understand to which extend and how you added noise to the spectra and treated it as a function of path length. So to me it is unclear what you refer to (for example in L. 207 the random noise at these separations") when you draw your conclusions on the detection limit.*

We have clarified the text in this paragraph to convey that we are comparing simulated differential absorption depth (HCHO signal) to calculated measurement noise in real spectra, i.e., no noise was added to the simulated spectra.

*Inconsistencies of numbers: At a couple of places your numbers are inconsistent. This might include rounding errors, but sometimes also wrongly calculated quotients of numbers. This of course leads to unnecessary ambiguities concerning the real results and I encourage you to fix that.*

We have fixed these inconsistencies in the body and conclusions.

*In a few points, your interpretation of your results conflicts with my personal understanding of the underlying physics. In these cases, I can see how your data might indicate your interpretations, but since it conflicts with my understanding of some fundamental principles I would ask you to consider some provided possible reasons for your differing results and, after that, if you still stand firm to your interpretation address these conflicts. The two most relevant examples for this are addressed in detail below in my remarks concerning L. 206ff and L. 359f.*

We believe we answer the concerns in these two most relevant examples below in a satisfactory manner while adhering to the same fundamental principles that the reviewer describes.

**Technical Corrections/Remarks**

*L. 10: Please refer to the general comment on detection limit as a function of path length.*

This comment is addressed above with General Remarks.

*L. 20: Please refer to the major point on Quantification of SNR dependency on path length.*

This comment is addressed above with Major Point.

*L. 62: it should be a translation of "up to ~6cm", since a central beam does not undergo a translation.*

Change made.

*L. 90f: Please refer again to the general comment on detection limit as a function of path length.*

Added "for typically used paths" on L92. The next sentence explains the breakdown in correlation when retroreflector overfilling becomes important.

*L. 98f: The 1 mrad beam divergence fits quite well to your data presented in Figure 6, so I assume it is true. But with the given values for aperture, focal length, and telescope reduction I arrive at 2.4 mrad. I don t know where the error lies, but maybe you can double check these numbers.*

This theoretical calculation the reviewer makes, $\frac{1}{2}\left(\frac{1}{9}\right)\left(\frac{3mm}{69mm}\right) \approx 2.4 mrad$, is correct, but not what we find in practice. This much beam divergence would lead to overfilling of a 24" array at just 63 m (one-way) compared to our field data of 150 m (one-way), which also matches the manufacturer's data. We have verified the system aperture, focal length and telescope reducing factor with the manufacturer (on several occasions) but the discrepancy remains. We have come to attribute this discrepancy to the fact that we are passing IR radiation through the telescope instead of visible radiation, which is what the telescope was designed for and what its documentation surely refers to, although we are not experts on astronomical telescopes.

*L. 102: Please refer to the general comment on cube corners shading themselves.*

This comment is addressed above with General Remarks.

*L. 123ff: Please refer to the general comment on the first report of retroreflector array size in open-path measurements.*

This comment is addressed above with General Remarks.

*L. 139: Why do you cite Rothman et al. (2013)? Did you use HITRAN 2012 and not the more recent versions? If so, why?*

Indeed, HITRAN 2012 was used, simply because time was short to test the practical effects of implementing spectroscopic updates across a range of species. It is the experience of the FTIR community (e.g., measurement networks like NDACC and TCCON) that HITRAN updates require a careful evaluation of new parameters. We are in the process of upgrading to the latest available version, which is 2020 (Gordon et al., 2022), which leads to small but significant fitting improvements for, e.g., $H_2O$. This was clarified in the paper.

*L. 157: While the difference between separation and total optical path length should be clear to every reader after the first mentioning, I would encourage you to stick either to optical path length or separation consistently in the paper when describing your setups.*

Good point. We have revised the paper throughout to refer more consistently to 'optical path length' (or 'OPL') when two-way path is meant and 'separation' when discussing the spectrometer-array separation; we cleared up a couple distance typos (300 m vs. 150 m) in the process, including in the abstract and Figure 3.

*L. 160: An interesting piece of information on the conduction of the experiment would be the timescale on which all of this happened. Did you move the array one increment roughly every 10 minutes, every hour, per day?*

The increasing path experiments were performed over the course of a few hours in 2015 and again in 2020. We added this clarifying information.

*L. 162: See comment on L. 157. I think you use separations and optical path here inconsistently yourself (separations, but two-way?).*

This comment has been addressed above (L157).

*L. 179: Concerning the exclusion of IR intensity <0.15 arb. Unit: Shouldn't this lead to a bias, since you would only keep the best measurements for long distances (where IR intensity is generally lower) but would include more measurements for shorter distances? I might just lack context here concerning your average IR intensity in these arbitrary units, since this might be a really low bar which filters mostly close to zero intensity spectra.*

As the reviewer suspects, this is a very low bar that removes a small percentage of the data, e.g., near-zero intensity spectra that occur on account of fog or rain in the open path, and which do not lead to meaningful retrieval results. We clarified this in the manuscript.

*L. 183: How representative is your metrological data if it is from 7 km away? If it is flat, the pressure should be fine, but could you give some context concerning the temperature?*

Ideally, meteorological data would be sourced from a sensor within the open path, however, this was not possible, so we sourced reliable data from the nearest available federal station, which is still coastal (not 7 km in land) and also at sea level. **We clarified this in the manuscript**.

Additionally, in other work we quantified the errors in retrieved concentrations due to perturbations in pressure and temperature as small (few %).

*L. 185: What parameters of "phase and shift" are you referring to? Maybe a "phase shift" in the parameterization of the instrument line shape (like often done for FTIR instruments) or a spectral shift?*

In this context "phase and shift" refer to two separate parameters, which correct for line shape asymmetry and line position shift, respectively. We clarified this in the manuscript.

*Figure 4: Your label of the color bar is inconsistent with your labels of the x and y axis, where you give you units in parentheses. I think it should be  Absorption(%)" then.*

Change made.

*L. 205: With your provided numbers of 0.06% and 0.015% it should be "~4x lower".*

This correction was made above with General Remarks.

*L. 206ff: I do not necessarily agree with your interpretation here. If you have a maximum absorption for HCHO of 0.12% you would be in a close to linear regime of Beer-Lambert's law. This means, that you would expect 5x the maximum absorption signal for 1500 m than for 300 m (you seem to observe x2). I could only explain this if there actually is an interfering species which significantly obscures the absorption features of HCHO, by reducing transmission significantly and bringing you in a way more saturated region of Beer-Lambert's law.*

Figure 5 (bottom panel) shows the differential absorption due to formaldehyde in a region including abundant interfering gases (H2O, CH4, and N2O, shown in the top panel; minimum transmittance due to H2O is not shown with these y-limits but is ~0.76). These interfering gases overlap heavily with the formaldehyde absorption features, exactly as per the reviewer's explanation of why we only observe <0.14% differential absorption at 1500 m as compared to at 300 m, when the distance is 5x higher at 1500 m (compared to 300 m) and the absorption should be 5x deeper in the linear case of HCHO.

*L. 224f, Figure 6: Please refer to the major point on Quantification of SNR dependency on path length.*

This comment is addressed in our response to the first paragraph of the reviewer's Major Point above. Model fitting was improved in Fig. 6 and the discussion was expanded.

*L. 229: Concerning your explanation of the low signal levels. This would mean a drop to 80% of the initial reflectivity of the cube corners on average, even including the 10% globar drop. And since you have a pristine array in the center, maximize for signal return and, thus, should only look at this pristine center for short distances, I find this difficult to believe. I do not consider the absolute value of your arbitrary signal fundamentally important to the results of your study, but this explanation seems lackluster. There are multiple settings for a Bruker Spectrometer which could change the level of the arbitrary signal, are you sure there is no better explanation?*

The reviewer questions the reasons we gave for a change in absolute signal levels between 2015 and 2020 while agreeing with us that they are fundamentally unimportant to our study. They argue that cube condition should not be a factor since at 100 m separation we should be looking at only the pristine cube cluster. This is not exactly the case (discussed in detail next), but we agree that there could be other factors that we are not accounting for, e.g., the precise orthogonal orientation of the retroreflector array to the IR beam, which is challenging to quantify and control on uneven terrain in the field. We *had* checked that major FTIR signal factors like pre-amp gains and aperture settings were the same and we made these clarifications in the paper.

Regarding which cubes are being viewed, with our 1 mrad beam divergence, at 100 m the 12" beam is already spread out to 20" (4 + 12 + 4) as compared to the 15" cube cluster (maximum horizontal width). By this argument alone (five inches corresponds to two cube widths) we could be looking at ~1/3 older cubes due to beam divergence at 100 m, and they would have to have degraded to 40% of initial reflectivity to give an area-average drop to 80% reflectivity (0.40*1/3 + 1.00*2/3 = 0.80). This *does* seem like too large a drop in cube reflectivity, so it's likely that orthogonal array alignment played a role in reducing 2020 signal levels. There is also some uncertainty in how the pan/tilt system, driven by precision stepper motors, causes the beam to traverse the array when we align the telescope on the retro in the x-y plane, but this uncertainty is small. In our system, each motor step is 0.005° = 0.087 mrad, which gives less than 1 cm of error at 100 m separation. Finally, the unavoidably irregular arrangement of cubes also played a role.

*L. 237: I do not understand what you mean with "outside of instrumental response".*

Noise was calculated from a spectral region that is not used to measure absorption lines/features due to being outside of the *detector's* response (but spectra are calculated for these wavelengths

and contain only instrumental noise; more on this at response below to L359f). The clarification between instrument and detector was made in the manuscript.

*L. 247: If the quoted number of -3.6/m stems from the fit in Figure 7 it should be rounded as -3.7/m.*

The reviewer is correct, however, Figure 7 has been changed as per a previous comment, and these numbers are no longer reported.

*L. 247, Figure 7: Please refer to the major point on Quantification of SNR dependency on path length.*

This comment is addressed above under Major Point. Model fitting was abandoned in Figure 7 and the discussion was modified.

*L. 256: "lager" should be "larger".*

Correction made.

*L. 257ff: I do not really see why you differentiate in this way between "a larger retroreflector array (and higher SNR)" and "different acquisition times". In the first order both (array size and acquisition time) just influence SNR, which then influences the retrieved concentration precision.*

The reviewer correctly notes that both retroreflector array size and measurement acquisition time impact the SNR. The reason for differentiating these two mechanisms is to isolate the SNR increase due to the larger reflector array alone. Without this differentiation, we would not be able to attribute the SNR increase seen in the 2020 experiment to the larger array. This has been reworded in the paper to clarify our meaning and purpose.

*L. 260ff: You probably also could just have averaged 2 interferograms or spectra for the 1 minute measurement mode (if each has half of the number of scans, which is actually the critical information) and performed the evaluation from there. This would make the two modes of operation comparable even if you have systematic differences/drifts between two consecutive measurements.*

The reviewer correctly notes an alternative method of accounting for the difference in measurement acquisition time, by combining two 1-minute measurements into one 2-minute

measurement (as there are indeed half the scans in a 1-minute measurement as compared to a 2-minute measurement). Ideally, we would have preferred to do that, but in practice this is cumbersome due to limitations on proprietary software functioning, therefore, we chose to account for the SNR change due to acquisition time based on theory in order to save time.

*L. 265, Figure 8: Why an "arbitrary time index"? Isn't this just something like "days since start or measurements" (ignoring the offset)?*

I

t is true that the Time Index is in fact consecutive days (apart from the offset), but the days are arbitrary in that they are not relative to the start of each campaign. We have changed the x-axis title to "Consecutive Measurement Days" and removed the offset, shifting the days to start at 1. We also removed the lines of best fit from Figure 8 since they are not meaningful and were never discussed in the paper.

*L. 269f: If the quality of measurements is doubled by the larger retroreflector, this means that your instrument is not shot noise limited. I don't know if this is typical for such MIR instruments, but in a shot noise limited case double the reflector surface should mean double the signal (in case of overfilling) and sqrt(2) larger noise, meaning sqrt(2) larger SNR. If your instrument noise is dominated by other sources (detector electronics/thermal noise maybe) than this behaviour would be expected. Could you comment on that?*

We agree that a shot noise-limited system would show a $\sqrt{2}$ increase in SNR instead of a 2x increase in SNR. While we are not noise experts, our understanding is that FTIR instruments operating in the MIR are *not* shot noise-limited (unlike in the UV and visible). In this case it is possible, as the reviewer notes, that doubling the signal doubles the SNR. However, in our 2021 experiment we increased the array size by 50%, and reported a corresponding signal increase of 50% (L290), so we need reasons why the SNR just about doubles (1.95) instead of increasing by a factor of 1.5 (like the signal increase). It seems unlikely that the detector noise was reduced. What is more likely is that the alignment between the telescope and the array was much better in 2021 as opposed to 2018, as we gained experience with our system. Another variable lowering the SNR in August (2018) as compared to March (2021) is the greater absolute humidity in the summer (over the ocean), which lowers the signal appreciably (see L359f below and Figure 12). We have included this discussion in the manuscript.

*L. 290: I do not understand how this higher intensity in 2021 is consistent with your finding in Figure 6 and Line 228, where your signal with the larger reflector array is suddenly smaller. Could you clarify that?*

Figure 6 and  L228 refer the maximum IR signal intensity at **100 m optical path:** ~0.95 at Franklyn (smaller, cleaner array in 2015) and ~0.69 at Otter (larger array with mixed clean and used cubes in 2020). The statement on L290 refers to a different set of experiments at **1120 m optical path:**  ~0.44 *on average* in 2018 (using the Franklyn array, but 3 years later) and ~0.69 *on average* in 2021 (using the Otter array, but 1 year later).  So we actually have the large array producing the ***same*** signal level in 2021 as compared to 2020, and this is curious because the separation in 2021 is more than 10x greater than in 2020.  Again, this is consistent with the orthogonal array alignment being treated very carefully in 2021 (at the start of a long-term field campaign) as compared to in 2020 (during a one-day field test of a larger array, where relative change was the goal).

***L. 307ff,**  even though there are four times fewer spectra in each hour in 2021": The here underlying assumption/interpretation is (in my opinion) wrong. Precision of the individual measurement results within an hour is a property of the spectra, which should be better for 2021 due to larger reflector array and longer measurement time (both increase spectral SNR). The number of measurements within an hour do not influence the precision of the individual measurements, but would only influence the precision of the mean over an hour - basically the difference between a standard deviation and the standard deviation of the mean which you mixed up here, I think.*

Agreed that we mixed up the standard deviation and the standard deviation of the mean with this statement, which we removed.  To clarify, we agree that the 'precision' of an *individual* measurement is a property of the spectrum and that it is not influenced by the number of measurements within the hour.  We think of this level of 'precision' as 'measurement error', which has many sources that can be lumped into, e.g., 'measurement noise' and 'retrieval error'.  We use 'precision' in our work to mean the spread in a repeatedly measured constant value (reasonable for a trace gas over one hour), which is a commonly accepted definition of 'measurement precision'.

*L. 309ff, Figure 9: The notion, that there is no discernible diurnal pattern in 2018 but in 2021 due to the difference in precision seems like a stretch to me. Also the 2018 data shows clear signals/drifts over the day on a similar magnitude. This might be caused by something else, but is still larger than the mentioned diurnal pattern in the 021 data. Attributing a lack of clear diurnal pattern to the precision does not seem correct. Furthermore, the visual comparison between the two panels is not really fair, since the upper panel shows a higher time resolution, resulting in more, but less precise measurement points.*

This comment is referring to both Figure 9 and Figure 10, but we take the general point that it may be a bit of a stretch to attribute the lack of a discernible diurnal pattern in 2018 due to a lower precision *alone*. We have revised the discussion of Figure 10 to acknowledge this.

*Figure 10: Concerning the textbox at the top of each panel: giving the not only month and days, but especially the year of your data might be more important, since this is what you actually use to differentiate the datasets and refer to them.*

This change was made and font sizes were increased.

*Figure 11: Text is way too small and hard to read, even on a digital device with appropriate zoom.*

Figure 11 carries a lot of information and has been somewhat degraded in the export to PDF for review purposes. We will submit a higher resolution rendering of this figure for final production in the case of publication perhaps the figure can have landscape orientation.

*L. 354f: Should "Fig. 10" be "Fig. 11" in both cases?*
Yes. This has been corrected in both cases.

**L. 359f:** *I do not agree with the statement, that (gaseous) water amount reduces SNR, unless in your spectroscopic window the total water absorption reduces the IR signal throughput (which I could not gather from your previous data in Figure 4, 5, and 6 for example). Rather, I assume that your calculated noise is inflated for the longer path due to systematic errors when fitting water lines. In the spectral window where you determine your noise (Fig. 7) are some water absorption features (even though no particular strong ones) but they might be strong enough so that for higher amounts of water the fit residual might reach the magnitude of the noise level. Or, if you are only taking the standard deviation from a polynomial background in this spectral region, the lines itself would cause an error once they are deep enough. I did not fully understand your process here. But I would encourage you to double check that this is no artefact of the way you calculate your noise value. Or maybe I misunderstood completely how you ended up at your conclusion.*

It is exactly as the reviewer suspects – total water absorption reduces the IR signal throughput, though this is not visible in Figs. 4 and 5. To clarify, while spectral fitting of target gases is performed in smaller windows, such as the one shown in Figs. 4 and 5 for HCHO, our system records broadband mid-IR spectra between 650 $cm^{-1}$ and 6,500 $cm^{-1}$ using a photoconductive MCT detector and a ZnSe beamsplitter (at the heart of the Michelson interferometer). More

precisely, it records interferograms, which are co-averaged and then (fast) Fourier transformed into mid-IR spectra ranging from 0 cm$^{-1}$ to 7,900 cm$^{-1}$ because of the Nyquist sampling limit within a Michelson interferometer. [The optical path difference is measured by counting the fringes of a HeNe laser (15,800 cm$^{-1}$), also made to pass through the Michelson, in parallel to the mid-IR signal (there is a separate detector and a small transmissive region on the ZnSe beamsplitter for the HeNe beam).] Thus, above 6,500 cm$^{-1}$ there is no signal (outside of the detector's range of response) but plenty of instrumental noise to work with, as shown in Figure 7 for 7640 cm$^{-1}$ – 7740 cm$^{-1}$.

Within the broadband spectral region where the detector responds, water vapour is saturated or near-saturated over hundreds of wavenumbers centered on 1600 cm$^{-1}$ (6.3 μm) and 3500 cm$^{-1}$ (2.9 μm) and does indeed reduce the returning IR signal at higher absolute humidity (e.g., summer vs. winter) and at longer path lengths. As the centerburst in the interferogram (peak interferometric signal at zero path difference) is reduced, the area under the entire spectrum (the FT of the interferogram) is also reduced. Thus, we have that as signal (the 100% transmission line) is reduced due to water, so is the SNR.

Regarding the accuracy of the noise calculation also mentioned in this paragraph, the reviewer considers an artificial inflation of the noise value due to the presence of water lines, but the reviewer states that they are not clear on our procedure to find the noise. They mention "systematic errors when fitting water lines" but also suggest "taking the standard deviation from a polynomial background." We do fit a polynomial between 7640 and 7740 cm-1 and use that polynomial to detrend whatever signal remains in this region, leaving behind only random noise because: 1) when we plot all such de-trended spectra there are no correlated absorption features in a visible inspection of the data 2) this region is outside of the detector's response and 3) we checked the HITRAN database and find that water absorption line strengths in this region are more than 3 orders of magnitude less than at 1600 cm$^{-1}$ and 3500 cm$^{-1}$ (above), even in the case of any remaining true detector response. To summarize, our **noise** is not inflated by water in the noise calculation region and the maximum **signal** in the region of detector spectral response is indeed reduced by the presence of water.

We have made tracked clarifications to improve our description of noise calculations.

*Figure 12: Why do you plot this as a function of relative humidity and not specific humidity or even total water column if you consider the strength of the absorption features the relevant cause?*

Figure 12 is a plot of retrieved mixing ratio of formaldehyde (in ppb) on the y-axis and retrieved mixing ratio of water vapour (in %) on the x-axis (it is not the relative humidity). The mixing ratio is proportional to the water column.

*L. 370ff: As mentioned above, I do not agree with the generalized formulation of path length being inversely correlated to the detection limit.*

We qualified "inversely correlates" by "at moderate path lengths", as per the discussion above in the first General Remark. The next sentence begins with "At sufficiently long path lengths, however…" A similar change was made in the manuscript body and abstract.

*L. 389ff: If this refers to the results from Figure 7, I think you would need to compare the relative drop in SNR, not the absolute ones. Since they are on the same order of magnitude, this detail results not in a dramatic difference, but for your numbers in Figure 7 it would then be pretty exactly a factor of 2 (slightly below).*

We have revised the wording in the conclusions (and abstract) to correspond to the changes made in the manuscript body.

*L. 424f: You say that there is an optimum array size and path length combination for each observation. What is it for the ones you discussed?*

It was the intention of this paragraph to discuss only general considerations. In practice, an 'optimum' implies that several nearby values of a variable were tested, which is very difficult in longer term field measurements focused on the gas concentrations (rather than solely the technique) where the retroreflector must be located on a certain rooftop, or across a harbour of a fixed width. We believe that we accurately summarized the specific conclusions from our work in the preceding paragraphs of our Summary & Conclusions, leaving space for generally valid remarks at the end.

*Summary and Conclusion in general: of course many points above apply to the respective parts in summary and conclusion where they are picked up again.*

Key modifications in the body have also been changed in the Abstract and Summary and Conclusions.

*References*

*André Merten, Jens Tschritter, and Ulrich Platt, "Design of differential optical absorption spectroscopy long-path telescopes based on fiber optics," Appl. Opt. 50*

*Citation: https://doi.org/10.5194/amt-2024-97-RC1*

---

## Author Comment (AC2)

**Format Key:**

*Reviewer comments in blue italics.*
Author responses in black.

**RC2**: Anonymous Referee #2, 23 Aug 2024

*General comments:*

*This paper describes a nice study to look at the impact of compensating for increasingly long pathlengths with larger arrays of retro-reflectors with the aim of improving the detection limits and precision of measurements of formaldehyde at typical ambient concentrations of 1ppb in the study area.*

*I have marked this as a minor revision but to address my concerns some extra analysis as well as discussion is required.*

We thank RC2 for their thorough reading of our submission and their thoughtful comments, which have improved our manuscript.

*The paper in its current form lacks some important discussion points, such as:*

1. *The data of formaldehyde retrievals include a large proportion of negative concentrations, but the paper does not discuss any possible retrieval strategies that could be applied to help fit the interfering gases and minimise these negative retrievals.*

   We have included a discussion of some retrieval strategies and checks (further down) that can be applied in order to minimize the occurrence of negative retrieval values. While it is part of ongoing work, we think that a retrieval optimization is out of scope for this paper, which is focused on path length and retroreflector size, while only characterizing the precision and (negative) bias of retrieved values. Optimizing the retrieval to minimize negatives is a non-trivial task. To our knowledge, one can start by examining: updated spectroscopic database parameters for the most prevalent interferer (water, but also other interferers) including temperature and pressure dependencies; correlations between the target gas and other retrieved parameters (interfering species, instrumental parameters, continuum parameters); retrievals that also include temperature as a retrieval target; multi-step retrieval approaches where water can be retrieved first and then constrained in the HCHO window; right up to even switching retrieval algorithms altogether to approaches that are mathematically constrained to return only positive values.

2.  *Regardless of whether a better retrieval strategy could be designed, the formaldehyde retrievals could be analysed to determine an actual limit of detection for each of the measurement set-ups described so that a time-series of measurements of ambient concentrations above this detection limit could be provided. This really is required (in my opinion) to make this manuscript publishable.*

The reviewer raises an interesting point that cuts to the heart of the strengths and weaknesses of FTIR spectroscopic concentration measurements:  they are quite precise and can nicely measure relative changes (like diurnal variations or pollution plumes), but their accuracy may be off by ~5% (e.g., Smith et al., 2011, AMT, for relatively high abundance $CO_2$, CO and $CH_4$); in the case of 'threshold' gases like formaldehyde (~1 ppb level, heavily overlapped by interfering species), these accuracy (systematic) issues may cause dips into negative concentration values in retrieval algorithms that are not mathematically constrained to yield positive values (like ours).  While negative concentrations are unphysical, they still convey information on relative change in the whole timeseries (positives and negatives).  Moreover, limiting reported values to those above a theoretically determined threshold (using only *random* noise), does not report on the bias (*systematic* or accuracy) problem.

Given this context, we quantified how the spectral signature of formaldehyde compares to typical *random* noise values in Section 3.1 and used that to determine a minimum path. While we can easily use our random noise value and the spectral signature at a given path to calculate the minimum detectable concentration of formaldehyde (it will be better than the 1 ppb assumed in the simulations of Figures 4 and 5), the negative bias problems are arising from *systematic* noise introduced by water vapour while overlapping the formaldehyde features (Figures 11 and 12), and we have no easy way to quantify that (Smith et al., 2011 used reference gas cells in a laboratory setting).  We agree that a time series of positive formaldehyde values is required for formaldehyde *process studies*, but we believe that it is not required in this paper, and that a presentation of negative values is in fact instructive to consider when making choices on path length and array size given beam divergence and interfering gases.

3.  *The discussion of the differential absorption appears to be discussed only as a %, whereas by my understanding MALT will fit the absorption area (not depth). Simply using a higher spectral resolution would provide a greater absorption depth and probably improve the detection limit and precision. Whilst this may not be possible with the equipment owned by the authors – it should be included in the discussion (since it will*

*also benefit the problem of interfering gases by improving the selectivity as the absorption lines become separated at higher resolutions).*

The reviewer is correct that MALT fits area and not depth. The instrument is operating at its maximum 'mid-range' resolution of 0.5 cm$^{-1}$. There is a limit to how much one can resolve the rotational structure in the horizontal open-path context because of strong pressure-induced line broadening processes, particularly for large molecules. We noted these points in Section 2.2.

*Specific comments:*

*I provide some more specific comments for some different sections of the paper below:*

*Abstract*

*Line 16: We demonstrate that two-way path lengths > ~300 m are necessary for robust HCHO spectral signatures (at typical random plus systematic noise levels." But this MUST also depend upon the ambient concentration!! Some discussion about typical concentrations of formaldehyde in different environments would help set this context.*

The reviewer is correct. We included the assumed concentration in the abstract and the range of ambient values in Section 2.1.

*Line 19: We demonstrate that the larger retroreflector array resulted in a smaller decrease in the signal-to-noise ratio as a function of measurement path, ~1.5 m-1 for the larger array as compared to ~3.6 m-1 for the smaller array" – Won t this depend upon the field of view of the individual spectrometer and telescope? If so then this is quite a specific detail that probably doesn t belong in an Abstract.*

Agreed. The Abstract and manuscript was revised significantly in this respect, also in response to RC1.

*Line 23 : (average standard deviation of 0.352 ppb for 2021 and 0.678 ppb for 2018 in hourly formaldehyde data bins over two days), - This is also detail that doesn t belong in an Abstract. And referring to the data by the different years is a strange choice – when it is the path length and cube-corner array size that it what matters not the date.*
*Quoting the standard deviation to 3 significant figures seems over the top.*

We replaced the reference to year with array size in the Abstract and we replaced the standard deviation values with a factor (~2x) for the precision increase.

**Experimental Design**

*Line 130 (p, T, precipitation, which causes IR beam extinction), Use of parenthesis is confusing here as the latter clause refers only to precipitation not temperature or pressure. Rephrase to clarify?*

Revised to "(p, T, and IR beam extinguishing precipitation)".

*Results*

*Line 229: Could the detector efficiency and/or pre-amp and amplifier gains also have decreased as the equipment aged? What is the difference in signal to noise at a part of the spectrum near to the formaldehyde absorption? Some of these factors will decrease the signal but not necessarily impact the signal to noise?*

We have revised this text to include detector aging as well, and the influence of alignment quality of the array in a plane orthogonal to the beam. It is difficult to estimate signal and noise accurately near the formaldehyde absorption because of highly variable interfering gases (there are no reliably 'free' regions of the spectrum near formaldehyde); we have clarified our discussion of how we estimate SNR instead.

*Figure 7: I am trying to understand the decision to calculate noise from a part of the spectrum outside of the detector response. I am not sure if this measure of signal to noise is the same as calculating it at a point in the spectrum with no absorption features but close to the wavenumber region where the gas of interest absorbs. Does this produce the same S:N? (Sorry if I am being slow – it has been a long week!) Maybe clarify this point in the text in any case??*

We have clarified the description of this calculation in the manuscript (also in response to RC1) and we also provided additional details why we think this is an equivalent method to assess noise within the response to RC1 (our response to their comment L359f). It is not possible to reliably check that both methods produce the same result because the spectrum is rather congested with interfering gases everywhere (e.g., the top panel of Figure 5 shows that water is everywhere present in the formaldehyde window). Before we settled on our approach, we attempted to calculate SNR from several apparently 'clean' spectral locations (including ones with full signal saturation) and found slightly differing results in all cases, with residual spectral structure after de-trending (which is absent outside of the detector response). However, in response also to RC1, we have modified the manuscript discussion to acknowledge that there is 'noise in the noise' and we abandoned fitting any trend lines to the decreasing SNR – while simply noting that SNR appears to decrease at a slower rate with the larger array.

*As well as the spectral signal to noise as discussed in and around Figure 7, there is the retrieval signal to noise. i.e. the spectrum to spectrum retrieved values give some idea of precision in a stable atmosphere. Given the values shown in Figure 9, I expect a discussion of the LOD here in terms of concentrations 3 x the retrieved value noise" – to determine where you have a clear detection of formaldehyde in the atmosphere.*

We quantify the measurement precision based on the scatter of retrieved values in Figure 9 and report how this varies in hourly bins in Figure 10. (These are the average standard deviations that we removed from the Abstract and replaced with a factor of 2 improvement (decrease) for the larger array). These would correspond to an LOD of ~0.35 ppb (large array) and ~0.69 ppb (small array) – *absent any systematic errors*, as we discussed in our response to General Comment 2 (RC2).

*Line 306 and 308: Are 3 and 4 significant figures justified here?*
We aimed for 3 decimal places, so 3 or 4 significant figures, which is admittedly high. We have changed values to 2 decimal places (2 or 3 sig figs), which is easier for a reader to absorb and compare (though also arbitrary).

**Citation**: https://doi.org/10.5194/amt-2024-97-RC2

---

## Author Response (AR2)

**Format Key**

Reviewer comments in blue italics

Author responses in black

**Reviewer 1: Anonymous reviewer #1, 09 Jan 2025Checklist for reviewers:1) Good2) Good3) GoodManuscript should be:Accepted subject to technical corrections**

**Suggestions for revision or reasons for rejection:**

In my opinion, the authors addressed all relevant points which were raised in the first round of discussions. While some of the details of how these points were addressed might be debatable, I do not see any major issues with the manuscript. Since these debatable details do not challenge the core message, findings, and results of the manuscript, I would be happy with some technical corrections or, if the handling editors considers this more appropriate, even a publication as is. Below, I provide a few comments, which do not need to be addressed necessarily and a few technical corrections for the authors to consider and I would be happy to leave all further discussions of the paper to the wider scientific community.

We thank Reviewer 1 for taking the time to read and provide valued feedback on this manuscript.

L. 170: I might be misreading this, but it is of course not the software which is operating at its maximum capacity – Bruker also uses this software for their high resolution instruments. The instrument simply operates at its (hardware limited) maximum optical path difference.

Indeed. We tweaked the wording to make it more clear that resolution is hardware-limited.

L. 180: Here maybe not necessary, but I generally like to also cite the 1977 erratum to the 1976 paper (appeared in J. Opt. Soc. Am., Vol. 67, No.3, March 1977, page 419) since the table with the quoted parameters change (which is typically the main reason somebody looks this up these days). Since this is often forgotten, even some textbooks have the table with the wrong values. An interesting point. A reference to the erratum has been added to the manuscript.

L. 189: It just occurred to me when reading this: overfilling should in principle also reduce the (effective) field of view (FOV) and with that the instrument line function if it is limited by the FOV. Especially in cases with strong interfering absorbers (like here) a slightly wrong ILS might cause such issues as the "negative" amounts of HCHO. But it is likely difficult to judge if this actually happens.

Indeed. As part of ongoing work, the effect of fitting or fixing the FOV in the retrieval continues to be investigated, including correlations between HCHO, FOV and other fitted parameters.

L. 240: It could have been interesting to leave the actually distance when overfitting occurs a free fitting parameter. But maybe you had to little data constraining your fit.

Indeed. Fitting without constraining the overfilling distance was unsuccessful in our sparse dataset.

**L. 301: Technically it should be a factor of sqrt(2)≈1.4 not 1.5.**

This paragraph is a bit dense because both array size and acquisition time were changed between experiments. The factor of sqrt(2) is a result of the change in acquisition time (applies to the SNR), while the factor of 1.5 results from increasing the retroreflector area by 50% (applies to the IR signal). We tweaked the wording to stress this better.

L. 302ff: I would not think that the signal level is that sensitive to the orthogonal alignment of the retro array – that is why we use them after all. The water vapor hypothesis sounds more reasonable to me, but difficult to judge with the provided data. But I think it is also okay if this question is not fully resolved.

We also thought that the orthogonal alignment of the retro array would not be a critical factor, but in our later experiment we discovered that it mattered a lot more than we expected (probably in connection with the very long separation). Since at least two things were changing at once (alignment precision, water vapour), it is not possible to separate them from this dataset.

**Reviewer 2: Anonymous reviewer #2, 17 Jan 2025**

| Checklist for reviewers: | 1) Fair                             | 2) Good | 3) Good |
|--------------------------|-------------------------------------|---------|---------|
| Manuscript should be:    | Accepted subject to minor revisions |         |         |

**Suggestions for revision or reasons for rejection:**

The study summarizes how the retrieval of formaldehyde (HCHO) measured by Open-Path Fourier Transform Infrared spectroscopy is influenced by different path lengths and retro reflector array sizes. Generally, the study can be divided into three parts, first of all spectral simulations for different open-path lengths were performed to determine the theoretical minimum path length for the detection of 1 ppb of formaldehyde given all other influencing variables are constant. Second, the impact of the retro reflector on the signal strength received at different path lengths has been analysed using data from two field campaigns. Finally, the retrieved formaldehyde concentrations for the two different array sizes at a fixed path length are evaluated, again using data from two field campaigns.

We thank Reviewer 2 for taking the time to thoroughly read and comment on the manuscript during this process.

**General comments:**

The manuscript is well structured and flows smoothly, making it easy to follow most of the time. Personally, I would suggest to add a table where the experiments throughout the years are listed and an indication of which of the two retro arrays and open path lengths were used, so the readers have central section to look up the details. Maybe also add the array size to the labels of the respective Figures or to the caption of each Figure.

This is a good suggestion and a table has been added to section 3.3 with a summary of the key acquisition data for each field experiment in this section. Figure captions were also updated to include information about array size (as 'large' or 'small').

Unfortunately, the measurements spanned over several years and the results are influenced not only by specifically introduced changes, but also by aging of the instruments IR light source and of the retro reflectors. Especially the improvement in Signal-to-Noise while using a larger reflector array could have been better highlighted by a more methodical experimental setup e.g., by changing the array size within one measurement campaign every day by covering parts of it for a few days. I acknowledge, that this will be time

**consuming and might not be feasible, but I think this would make the results easier to compare.**

We agree that multiple factors (including wear and tear on individual retro cubes) changing together is not desirable for strict experimental control. Indeed, it is not feasible to repeat the extended field work, which was designed with a scientific (and not technical) purpose in mind. We do have one very short set of spectra (minutes) with a part of the array covered and (as expected) the array with all cubes exposed leads to higher SNR spectra.

**Nevertheless, the manuscript nicely shows how a larger reflector array can beneficial for larger open path lengths.**

Thanks. The one-day experiments with variable path and increasing overfilling are most controlled in this regard.

**Specific comments:**

Lines 274ff: This contradicts the information presented in Figure 7 at the bottom, where only one datapoint of the "Otter lake" experiment shows a better SNR when the retro array is overfilled. Even though it is mentioned that the SNR is influenced by non-constant noise and that interpretation should be approached with caution. Maybe add an additional third panel to Figure 7, where the SNRs are normalized by the mean or median of the SNR of the underfilled retro array for each respective experiment. This would show the relative decrease of SNR caused by different path lengths and should indicate a slower decrease for the larger retro reflector array.

We highlighted the relative decrease of SNR in the original submitted manuscript, including lines of best fit, but these were removed in the revised manuscript because of the uncertainty due to non-constant noise. Dividing by the mean or median is an interesting suggestion, but each experiment has a different range of SNR values, hence showing such a normalization of the SNR could be misleading in its own way. In the end, having the reader infer (i.e., 'eyeball') the slower SNR decrease with a larger array (Otter Lake) seems like the most non-complicating presentation of this dataset.

**Line 461ff: Would it be possible to add a small section about what you found the best compromise in your study?**

A sentence has been added to the very end of the manuscript to summarize our findings on the optimum OPL for HCHO in our coastal environment.

Technical corrections:

The array sizes in Figure 1 and in the text are given in inches and I am not sure whether this fits with the AMT guidelines, which strictly state SI or SI derived units should be used. Figure 8: The description still contains a half sentence of an earlier manuscript version, this should be removed.

All instances of inches have been replaced with centimeters. Figure 8 caption fixed.

Reviewer 3: Paton-Walsh, Clare, 18 Jan 2025

| Checklist for reviewers: | 1) Good    | 2) Good        | 3) Good           |
|--------------------------|------------|----------------|-------------------|
| Manuscript should be:    | Accepted s | ubject to tech | nical corrections |

**Suggestions for revision or reasons for rejection:**

I recommend that the article is accepted with a very minor tweak - which is to add a sentence or phrase at around line 15 of the Abstract stating upfront that formaldehyde is used as an example of a trace gas that has relatively weak absorption features at ambient concentrations and so is a challenge to retireve from OP-FTIR and therefore is sensitive to changes in the instrumental performance such as the path-length, beam divergence and array size.

We thank Dr. Paton-Walsh for her feedback for the improvement of this manuscript. A sentence has been added to the abstract of the manuscript to reflect that HCHO also serves as a proxy for any low abundance trace gas targeted for an OP- FTIR retrieval.